# KBQA-R1: Reinforcing Large Language Models for Knowledge Base Question Answering

**Xin Sun** [1] **Zhongqi Chen** [2] **Xing Zheng** [2] **Bowen Song** [2] **Qiang Liu** [3] **Shu Wu** [3] **Zilei Wang** [1] **Weiqiang Wang** [2] **Liang Wang** [3]

## Abstract

Knowledge Base Question Answering (KBQA) challenges models to bridge the gap between natural language and strict knowledge graph schemas by generating executable logical forms. While Large Language Models (LLMs) have advanced this field, current approaches often struggle with a dichotomy of failure: they either generate hallucinated queries without verifying schema existence or exhibit rigid, template-based reasoning that mimics synthesized traces without true comprehension of the environment. To address these limitations, we present **KBQA-R1**, a framework that shifts the paradigm from text imitation to interaction optimization via Reinforcement Learning. Treating KBQA as a multi-turn decision process, our model learns to autonomously navigate the knowledge base using a structured action space, refining its reasoning strategies based on concrete execution feedback rather than static supervision. Furthermore, we introduce Referenced Rejection Sampling (RRS), a data synthesis method that resolves cold-start challenges by strictly aligning reasoning traces with ground-truth action sequences. Extensive experiments on WebQSP, GrailQA, and GraphQuestions demonstrate that KBQA-R1 achieves state-of-the-art performance. Code and project page are available at https://github.com/sunxin000/KBQA-R1 and https://sunxin000.github.io/KBQA-R1/.

[1]University of Science and Technology of China, Hefei, China [2]Independent [3]Institute of Automation, Chinese Academy of Sciences, Beijing, China. Correspondence to: Bowen Song <wdboou@gmail.com>, Shu Wu <shu.wu@nlpr.ia.ac.cn>.

*Proceedings of the $43^{rd}$ International Conference on Machine Learning*, Seoul, South Korea. PMLR 306, 2026. Copyright 2026 by the author(s).

## 1. Introduction

Knowledge Base Question Answering (KBQA) aims to answer natural language questions by retrieving facts from large-scale Knowledge Bases (KBs) such as Freebase and Wikidata. Unlike Retrieval-Augmented Generation (RAG), which augments Large Language Models (LLMs) with unstructured text snippets, KBQA requires the model to produce executable logical forms (e.g., SPARQL or S-Expressions) that precisely navigate a KB schema. Here, *schema elements* refer to the exact relation names, entity types, and attributes that define the graph vocabulary (e.g., `film.actor.film`). This makes KBQA a stringent reasoning task: a model must understand the question, choose schema-consistent relations, and compose multi-hop queries without violating the executor's syntax or semantics.

Recent LLM-based KBQA systems have made substantial progress, but three limitations remain. **First**, end-to-end approaches such as KB-BINDER (Li et al., 2023), KB-Coder (Nie et al., 2024), and ChatKBQA (Luo et al., 2024) generate an entire logical form in one pass. They are efficient, but they cannot verify schema elements during generation, which often leads to *schema hallucinations*: executable-looking queries that mention invalid or irrelevant relations. **Second**, prompting-based step-by-step approaches (Sun et al., 2023; Luo et al., 2023) decompose reasoning into intermediate graph-exploration steps, but they rely heavily on in-context heuristics and strong commercial APIs rather than learning a task-specific policy for KB navigation. **Third**, supervised or search-augmented agentic approaches (Luo et al., 2025c) improve executability with synthesized traces or test-time search, but their traces can become *template-driven* action announcements rather than genuine analysis of KB feedback, and search-based inference adds substantial overhead. Recent graph-retrieval and GraphRAG methods (Mavromatis & Karypis, 2025; Li et al., 2024b;a) reduce hallucination by retrieving or decoding over local subgraphs, yet they still depend on offline subgraph construction or retrieval pipelines whose policies are not directly optimized for online KB interaction.

We present **KBQA-R1**, an action-centric reinforcement learning framework that turns KBQA into a closed-loop

interaction problem. Instead of generating raw query code in one shot, KBQA-R1 operates over a compact, typed action space, where actions such as `Find_Relation` and `Merge` are executed against the KB and converted into verifiable S-Expression fragments. At each turn, the model emits reasoning and actions, receives concrete KB feedback, and updates its trajectory until it produces a final answer. By optimizing the policy with outcome-oriented reinforcement learning rather than static imitation alone, KBQA-R1 encourages adaptive reasoning that interprets observations and justifies action choices based on executable feedback.

To make this reinforcement learning process stable, we introduce **Referenced Rejection Sampling (RRS)** as a warm-start data synthesis strategy. Standard rejection sampling from raw prompts is inefficient for KBQA because most zero-shot trajectories contain malformed actions, hallucinated relations, or invalid query structures. RRS addresses this cold-start problem by conditioning trajectory generation on a reference sequence of ground-truth actions while requiring the model to reconstruct a coherent reasoning trace around those executable steps. Accepted trajectories are then stripped of reference hints before supervised fine-tuning, so the policy learns KB-grounded reasoning patterns without depending on hidden guidance at inference time.

Our main contributions are summarized as follows:

- We propose **KBQA-R1**, a multi-turn reinforcement learning framework that grounds LLM reasoning in verifiable KB actions, enabling closed-loop interaction with the knowledge base.

- We introduce **Referenced Rejection Sampling (RRS)**, a data synthesis strategy that aligns reasoning traces with ground-truth action sequences and mitigates hallucinated logic during warm-start training.

- Extensive experiments on WebQSP, GrailQA, and GraphQuestions demonstrate that KBQA-R1 achieves state-of-the-art performance. Moreover, RL training enables precise navigation on the knowledge graph: KBQA-R1 requires over 70% fewer LLM calls than GPT-4-based prompting methods (ToG, PoG), while achieving superior accuracy with only a Llama-3.1-8B backbone.

## 2. Related Work

KBQA has been studied through semantic parsing and retrieval-based methods, and recent LLM-based systems mainly fall into three lines: end-to-end logical-form generation (Li et al., 2023; Nie et al., 2024; Luo et al., 2024; Jiang et al., 2023), agentic graph exploration with interleaved reasoning and tool feedback (Gu et al., 2023b; Huang et al., 2024; Sun et al., 2023; Luo et al., 2023; Xiong et al., 2024;

Chen et al., 2024; Jiang et al., 2024), and search- or graph-augmented reasoning methods that use MCTS, GraphRAG, or local subgraph retrieval to improve robustness (Luo et al., 2025c; Mavromatis & Karypis, 2025; Li et al., 2024b;a; Luo et al., 2025a). These approaches reduce schema hallucination in different ways, but they either rely on one-shot generation, test-time prompting/search, or external retrieval pipelines whose policies are not directly optimized for executable KB interaction. KBQA-R1 instead trains a single policy to act in a typed KB environment with verifiable feedback, so the model internalizes schema-aware navigation during training rather than relying on heuristic exploration at inference time. We provide a more detailed comparison with these lines of work in Appendix A.

## 3. Preliminaries

**Knowledge Base and Executor.** We consider a knowledge base (KB) as a directed multi-relational graph $\mathcal{K} = (\mathcal{E}, \mathcal{R}, \mathcal{F})$, where $\mathcal{E}$ is the set of entities, $\mathcal{R}$ is the set of relations, and $\mathcal{F}$ is the set of factual triples. Each triple $f \in \mathcal{F}$ has the form $(h, r, t)$ with head entity $h \in \mathcal{E}$, relation $r \in \mathcal{R}$, and tail entity $t \in \mathcal{E}$. An executor $\mathcal{E}$ (e.g., a SPARQL endpoint) takes a structured query over $\mathcal{K}$ and returns an answer set, which serves as the environment feedback in our framework.

**KBQA Task.** Given a natural language question $q$, the KB $\mathcal{K}$, and a set of topic entities $E_q \subseteq \mathcal{E}$ mentioned in $q$, the goal of Knowledge Base Question Answering (KBQA) is to produce an answer set $\mathcal{A}_q \subseteq \mathcal{E}$ that correctly responds to the question. Following the standard KGQA setting used by prior systems such as ToG, RoG, PoG, GNN-RAG, SubgraphRAG, and KBQA-o1 (Sun et al., 2023; Luo et al., 2023; Chen et al., 2024; Mavromatis & Karypis, 2025; Li et al., 2024b; Luo et al., 2025c), we assume that entity mentions in $q$ are already linked to the KB and the corresponding topic entities $E_q$ are given as input. In classic semantic-parsing based KBQA, this task is realized by generating a logical form (e.g., SPARQL or S-Expression) in one shot and executing it against the KB. In contrast, our framework rephrases the task as learning a multi-step interaction policy.

**Agentic KBQA as Sequential Decision Making.** In our framework, we view the large language model as a stochastic policy $\pi_\theta$ that interacts with the KB environment via a compact, validated action space. At each step $t$, the agent observes a context $c_t$ summarizing the dialogue history, including prior reasoning (`<think>` blocks), actions (`<action>` blocks), and tool feedback (`<information>` blocks). Conditioned on $c_t$ and the original question $q$, the policy samples an action $a_t$:

$$a_t \sim \pi_\theta(\cdot \mid q, c_t).$$

The action $a_t$ is grounded into an S-Expression fragment and

executed by the executor $\mathcal{E}$ over $\mathcal{K}$, yielding an observation $o_t$ (e.g., retrieved entities or diagnostic messages). The triple $(c_t, a_t, o_t)$ is appended to the trajectory, and the context is updated accordingly. This interactive loop continues until the agent outputs a final answer $\hat{\mathcal{A}}_q$ or a maximum number of steps $T$ is reached. We denote a complete trajectory by $\tau = \{(c_1, a_1, o_1), \ldots, (c_T, a_T, o_T)\}$.

# 4. Method: The KBQA-R1 Framework

## 4.1. Prompt and System Workflow

Our system is a multi-turn agent system inspired by the ReAct paradigm (Yao et al., 2022). At each turn, the LLM emits one or more actions to interact with the KB environment, and the environment returns the corresponding observations, forming the Think-Action-Information loop shown in Figure 1. After multiple turns of KB exploration, the model outputs the final answer. Algorithm 2 in Appendix B.8 gives the formal execution loop, and Appendix B.9 provides a complete step-by-step interaction trace.

### 4.1.1. PROMPTING TEMPLATE FOR ACTION-BASED REASONING

We use a structured, action-based prompting format to support multi-turn interaction with the Freebase executor. At each turn, the model produces (i) a rationale in <think>...</think>, (ii) one or more executable KB exploration actions in <action>...</action>, and is then conditioned on the environment feedback returned in <information>...</information>. The loop repeats until the model outputs the final answer in <answer>...</answer>. Table 2 summarizes the prompt template.

### 4.1.2. ACTION SPACE

Prior semantic parsing approaches to KBQA (Gu & Su, 2022; Luo et al., 2024; Ye et al., 2022) typically require the model to emit a full, nested S-expression in a single pass. This design is notoriously brittle: a single token-level error (e.g., a typo in a relation name or a mismatched parenthesis) can render the entire program unexecutable and cause the query to fail.

Following the recent KBQA-o1 framework (Luo et al., 2025c), we instead adopt a compact, discrete action space that decomposes logical-form construction into a sequence of simple, verifiable steps. Concretely, each action corresponds to an atomic operation over the evolving logical expression, such as extending from an entity along a relation (Find_relation), intersecting two partial expressions (Merge), or applying aggregation and comparison operators (Order, Compare, Count, Time_constraint).

As summarized in Table 1, every action is defined by (i) its arguments, (ii) a target functional update on the current expression (e.g., JOIN, AND, ARG, CMP, TC, COUNT), and (iii) the corresponding S-expression template.

Actions are converted into an S-Expression list, then translated into SPARQL queries (Pérez et al., 2009) executed against the KB. The resulting observations are appended to the dialogue state visible to the model.

### 4.1.3. RELATION RETRIEVAL AND CONFIDENCE GATING

LLM-proposed relations can be noisy or ambiguous due to the well-known hallucination problem (Zhang et al., 2023b). To mitigate this, we introduce the **Relation Retrieval and Confidence Gating** (RRCG) module. The RRCG module acts as a validation layer, verifying the agent's proposed textual relation before execution.

Let $r_{\text{agent}}$ be the original textual relation proposed by the agent for the current entity $e_c$. Let $R(e_c)$ be the set of all neighboring schema relations of $e_c$ in the knowledge base. The core of the RRCG module is a similarity function $Sim(\cdot, \cdot)$, implemented using dense retrieval techniques (Karpukhin et al., 2020; Zhao et al., 2024), which scores $r_{\text{agent}}$ against every schema relation $r_s \in R(e_c)$. We define $s_{\max} = \max_{r_s \in R(e_c)} Sim(r_{\text{agent}}, r_s)$ as the highest similarity score, with $r_s^* = \arg\max_{r_s \in R(e_c)} Sim(r_{\text{agent}}, r_s)$ being the best-matching schema relation.

Based on $s_{\max}$ and predefined thresholds $\tau_{\text{high}}$ and $\tau_{\text{low}}$ (where $\tau_{\text{high}} > \tau_{\text{low}}$), the action is categorized into three confidence tiers: (1) **Auto-Validation**: If $s_{\max} \geq \tau_{\text{high}}$, the action is directly executed using $r_s^*$. (2) **Tentative Acceptance**: If $\tau_{\text{low}} \leq s_{\max} < \tau_{\text{high}}$, the action proceeds but the observation includes top-$k$ candidates to signal uncertainty. (3) **Rejection**: If $s_{\max} < \tau_{\text{low}}$, the action is marked invalid and a diagnostic message with neighboring relations is returned to guide re-selection.

## 4.2. Rejection Sampling and Supervised Fine-Tuning Warm-Start

To effectively warm-start the policy before RL and resolve the "cold start" problem, we propose Referenced Rejection Sampling (RRS), a data synthesis strategy that grounds the model's reasoning in verifiable execution steps. The key insight behind RRS is that successful KBQA trajectories must align with executable action sequences. Standard rejection sampling (Yuan et al., 2023) from raw prompts suffers from very low acceptance rates due to the task complexity and the base LLM's weak zero-shot ability on structured KB queries. Simply increasing sampling temperature or budget yields diminishing returns, as most generated trajectories

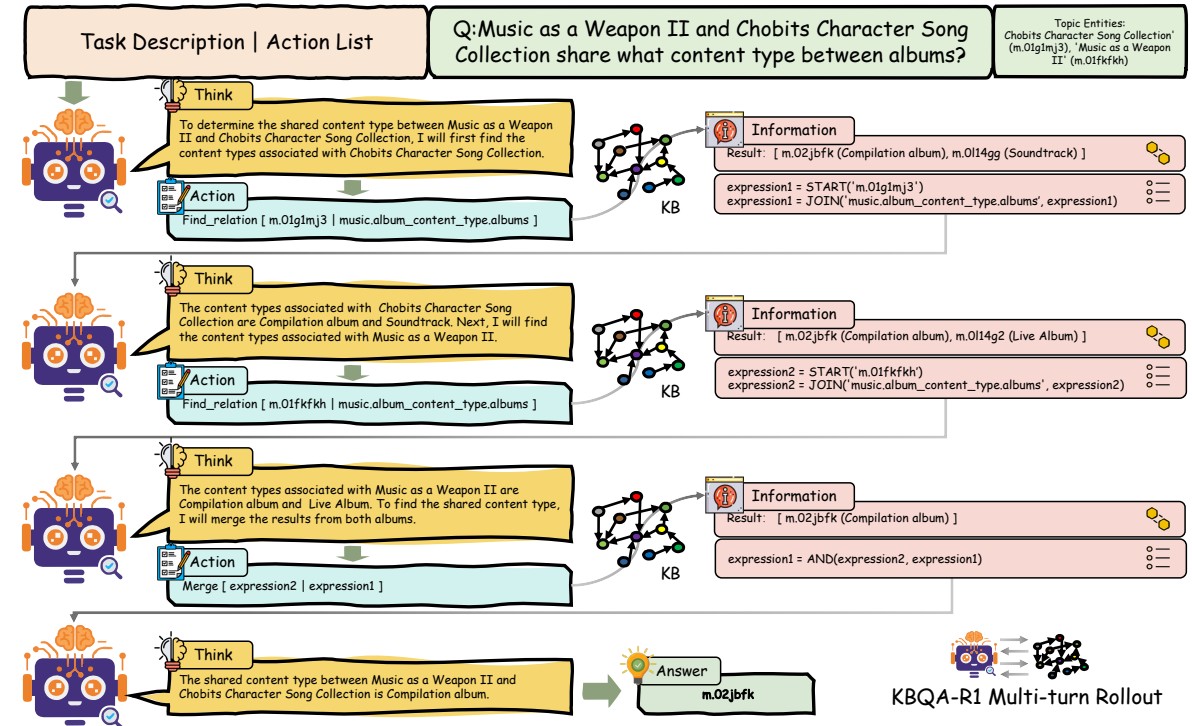

*Figure 1.* Overview of KBQA-R1. The agent iteratively executes a *Think-Action-Information* loop, selecting atomic actions and receiving KB feedback.

| Action | Arguments | Target Function | Equivalent Logical Form |
|---|---|---|---|
| Find_relation | *entity* │ *relation* | *expression* = JOIN('*relation*', START(*entity*)) | (JOIN *relation entity*) |
| Merge | *expression1* │ *expression* | *expression* = AND(*expression1*, *expression*) | (AND (*expression1*) (*expression*)) |
| Order | *MAX/MIN* │ *expression* │ *relation* | *expression* = ARG('*mode*', *expression*, '*relation*') | (*mode* (*expression*) *relation*) |
| Compare | *le/lt/ge/gt* │ *relation* │ *number* | *expression* = CMP('*mode*', '*relation*', *number*, *expression*) | (*mode relation number* (*expression*)) |
| Time_constraint | *relation* │ *time* | *expression* = TC(*expression*, '*relation*', '*time*') | (TC (*expression*) *relation time*) |
| Count | *expression* | *expression* = COUNT(*expression*) | (COUNT (*expression*)) |

*Table 1.* Action space of KBQA-R1.

*Table 2.* Action-based reasoning prompt template for KBQA-R1.

You are an expert assistant for querying the Freebase knowledge base using structured reasoning actions.
Answer the given question about Freebase knowledge base.
Conduct reasoning inside `<think>...</think>` before emitting actions.
Provide structured actions inside `<action>...</action>`.
The system returns query results between
`<information>...</information>`.
Return the final answer inside `<answer>...</answer>` using MIDs or literal values.
Available Actions: `{Action Descriptions}`; Question:
`{QUESTION}`.

contain hallucinated relations or malformed S-Expressions.

RRS addresses this by providing the model with a *reference action sequence* as implicit guidance during generation. This constraint forces the model to: ❶ **Ground reasoning in execution:** The model must justify *why* each reference action leads toward the correct answer, rather than fabricating post-hoc explanations. ❷ **Learn action-observation correspondence:** By observing the actual KB feedback for each

ground-truth action, the model internalizes the mapping between actions and their environmental consequences.

### 4.2.1. RRS Pipeline

Given a training example $(q, \mathcal{A}, S^*)$ where $q$ is the question, $\mathcal{A}$ is the gold answer set, and $S^*$ is the gold S-Expression, the RRS pipeline proceeds as follows:

**Step 1: Action Extraction.** Parse $S^*$ to extract the ground-truth action sequence $\mathbf{a}^* = (a_1^*, a_2^*, \ldots, a_k^*)$, where each $a_i^*$ corresponds to an atomic operation (e.g., Find_relation, Merge).

**Step 2: Referenced Rollout.** Execute a rollout where the model generates reasoning traces (`<think>`) conditioned on observing the reference actions. At each step $t$, the prompt includes the next ground-truth action $a_t^*$ as a reference.

**Step 3: Trajectory Filtering.** Accept trajectories that (a)

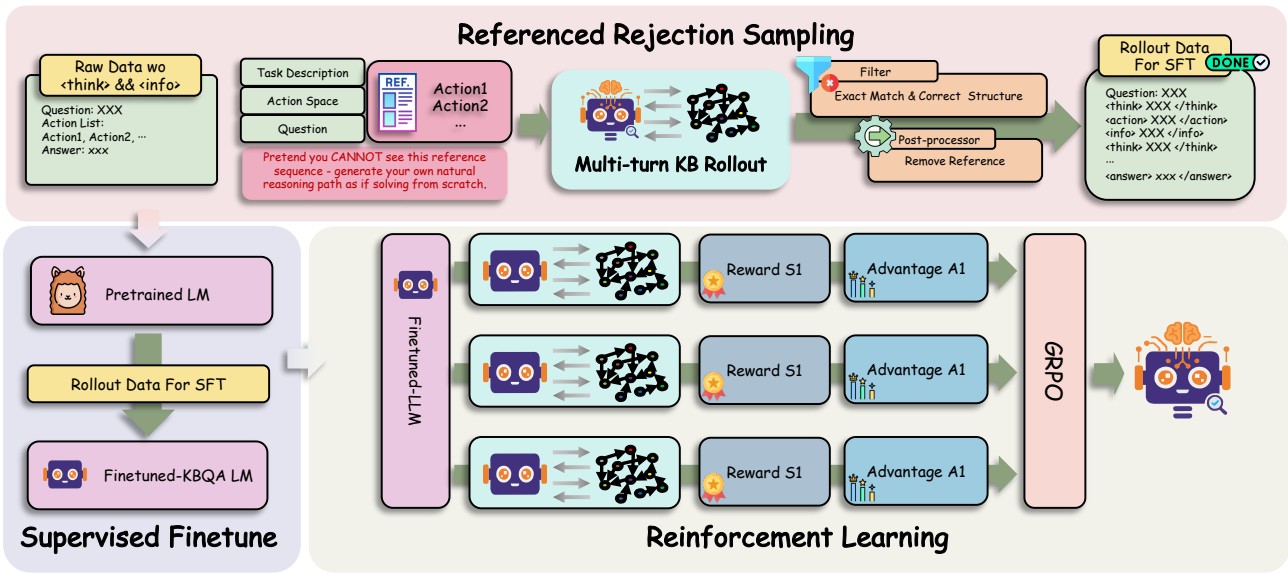

*Figure 2.* Two-stage training pipeline. **Stage 1:** Referenced Rejection Sampling generates trajectories conditioned on gold action sequences, filtered by correctness for SFT. **Stage 2:** GRPO optimizes the policy via outcome-based rewards from multi-turn KB rollouts.

successfully reach the correct answer with $\text{F1}(\hat{\mathcal{A}}, \mathcal{A}) \geq \tau$, and (b) maintain correct structure format of the tags.

**Step 4: Reference Stripping.** Before adding accepted trajectories to the SFT dataset, we strip all reference hints from the prompts. This ensures the model learns to reason independently at inference time.

The resulting SFT dataset $S_{\text{RRS}}$ contains high-quality trajectories where each reasoning step is grounded in verifiable KB interactions. This approach achieves significantly higher acceptance rates compared to raw rejection sampling while producing more robust reasoning patterns.

### 4.2.2. SFT TRAINING

Each turn in the accepted trajectory is converted into an independent training sample, where the context (including prior turns and KB feedback) serves as input and the model's response serves as the target. The SFT loss is computed only on response tokens. The resulting checkpoint initializes the policy for GRPO optimization.

### 4.3. Reinforcement Learning Optimization

The policy is further refined via Reinforcement Learning (Kaelbling et al., 1996), optimizing a composite reward signal using our GRPO algorithm (Shao et al., 2024).

### 4.3.1. REWARD FORMULATION

We define a composite reward $R$ to guide the policy, composed of three main components: an outcome reward ($r_{\text{outcome}}$), a format reward ($r_{\text{format}}$). The primary component

is $r_{\text{outcome}}$, which measures the factual accuracy of the final answer. To make this signal robust against annotation variations, it is calculated as the *F1* score between the predicted answers $\hat{\mathcal{A}}$ and all available gold answer variants $\mathcal{A}$ for a given prompt. The second component is the $r_{\text{format}}$. This provides a bonus based on desirable structural properties, such as tag completeness and correct tag order. Crucially, this reward is applied *only when $r_{outcome} > 0$*, ensuring the agent is not rewarded for good syntax when the answer is completely wrong.

The total reward $R$ for a trajectory is the weighted sum of these components, where $\mathbb{I}[\cdot]$ is the indicator function:

$$R = \lambda_{\text{outcome}} \cdot r_{\text{outcome}} + \lambda_{\text{format}} \cdot \mathbb{I}[r_{\text{outcome}} > 0] \cdot r_{\text{format}} \quad (1)$$

### 4.3.2. POLICY OPTIMIZATION (GRPO)

We optimize the policy $\pi_\theta$ using GRPO (Shao et al., 2024). The overall objective maximizes the expected clipped advantage, regularized by a KL-divergence term against a frozen reference policy $\pi_{\text{ref}}$ to ensure training stability (Ouyang et al., 2022):

$$\max_\theta \mathbb{E}\Big[ \min\big(r_t \hat{A}_t, \text{clip}(r_t, 1-\epsilon, 1+\epsilon)\hat{A}_t\big)\Big]$$
$$- \beta \, D_{\text{KL}}[\pi_\theta \| \pi_{\text{ref}}] \quad (2)$$

where $r_t = \pi_\theta(a_t|s_t)/\pi_{\theta_{\text{old}}}(a_t|s_t)$ is the importance sampling ratio, $\epsilon$ is the clipping threshold, and $\beta$ controls the KL penalty strength.

The key feature of GRPO is its definition of the advantage function $\hat{A}_t$. For a given prompt $x$, we execute $n$ rollouts

with the current policy $\pi_\theta$ to generate $n$ candidate trajectories $\{y_i\}_{i=1}^n$ and their corresponding scalar rewards $\{r_i\}_{i=1}^n$. Instead of using a learned value function (as in standard actor-critic methods (Schulman et al., 2015)), GRPO computes the advantage by centering the rewards within this group, using the group's mean reward as a baseline:

$$\hat{A}_i = r_i - \frac{1}{n} \sum_{j=1}^n r_j \tag{3}$$

# 5. Experiments

## 5.1. Experimental Setup

### 5.1.1. DATASETS

We conduct experiments on three widely-used KBQA benchmarks grounded on Freebase (Bollacker et al., 2008): GrailQA (Gu et al., 2021), WebQSP (Yih et al., 2016), and GraphQuestions (Su et al., 2016). Following the experimental setup of KBQA-o1 (Luo et al., 2025c), we use the official splits and evaluation protocols (with GrailQA evaluated on the dev set as in prior work). Detailed dataset statistics and split sizes are provided in Appendix B.1.

### 5.1.2. BASELINES

We compare KBQA-R1 with representative KBQA baselines, including (i) `fine-tune-based` methods trained with full supervision: RnG-KBQA (Ye et al., 2022), DecAF (Yu et al., 2022), TIARA (Shu et al., 2022), SPARQA (Sun et al., 2020), BERT+Ranking (Gu et al., 2021), ArcaneQA (Gu & Su, 2022), and KBQA-o1 (Luo et al., 2025c); (ii) `prompting-based` methods: KB-BINDER (Li et al., 2023), KB-Coder (Nie et al., 2024), and ARG-KBQA (Tian et al., 2024); and (iii) recent graph-retrieval and agentic reasoning methods such as RoG (Luo et al., 2023), GNN-RAG (Mavromatis & Karypis, 2025), and SubgraphRAG (Li et al., 2024b). Detailed baseline descriptions are provided in Appendix B.2.

### 5.1.3. EVALUATION METRICS

We evaluate all methods using standard KBQA metrics: **Exact Match (EM)** measures the percentage of questions where the predicted answer set exactly matches the gold answer set. **F1 Score** computes the harmonic mean of precision and recall at the entity level, providing a more lenient measure that accounts for partial correctness.

### 5.1.4. TRAINING SETUP

We use Llama-3.1-8B-Instruct as the default backbone. We run RRS with a stronger instruction-following backbone (Qwen-2.5-72B-Instruct) to obtain high-quality trajectories, and then distill these trajectories into our Llama-3.1-8B-Instruct policy via supervised fine-tuning. Full hyperparameters (e.g., epochs, learning-rate schedules, batch sizes, KL coefficient $\beta$, and clipping ratios) as well as infrastructure details are reported in Appendix B.3.

## 5.2. Main Results Analysis

For GrailQA dataset (Table 3), KBQA-R1 delivers consistent gains over the strongest fine-tuned baseline KBQA-o1 across all three generalization levels. In the i.i.d. split, KBQA-R1 improves EM by about $+12\%$ and F1 by roughly $+6\%$. In the compositional split, which stresses recombining seen schema elements, KBQA-R1 still achieves a solid margin of around $+5\%$ F1. **Most notably, in the zero-shot setting—where relations and compositions are unseen during training—KBQA-R1 boosts EM by more than $+15\%$ and F1 by about $+9\%$ over KBQA-o1. Overall on GrailQA, these improvements translate into gains of roughly $+8\%$ F1 and $+12\%$ EM, highlighting that execution-grounded reinforcement learning significantly enhances out-of-distribution generalization rather than merely fitting the training distribution.** On WebQSP (Table 4), KBQA-R1 attains 83.4% F1, outperforming graph-retrieval methods such as RoG, GNN-RAG, and SubgraphRAG, while exceeding fine-tuned systems such as TIARA and DecAF. Compared with the Llama-3.1-8B-based MCTS-KBQA, KBQA-R1 achieves about $+7\%$ absolute F1 improvement, suggesting that learned policies are more effective than MCTS search heuristics under the same backbone. On GraphQuestions (Table 5), which emphasizes long multi-hop queries, KBQA-R1 yields around $+5\%$ absolute F1 gain over KBQA-o1 and consistently surpasses earlier graph-based methods such as CoTKR and ArcaneQA. These results indicate that KBQA-R1 effectively enhances reasoning capabilities across diverse KBQA challenges, including complex multi-hop queries.

## 5.3. Ablation Study

We conduct ablation studies to quantify the contribution of the key components introduced in Section 4, including RRCG, the structured action space, the RRS warm-start, and GRPO-based RL optimization.

**Agent Architecture Ablations.** The most significant performance drops occur when removing core architectural components. (1) *w/o RRCG* results in an average F1 drop of about 18%, with GrailQA suffering the largest degradation ($-19.0\%$). Without relation retrieval and confidence gating, the agent must rely solely on the LLM's parametric knowledge to select relations, leading to frequent hallucinations on unseen schema elements. The impact is particularly severe on GraphQ ($-16.1\%$), where complex multi-hop queries require precise relation grounding. (2) *w/o Multi-turn* causes the most dramatic decline (about $-25\%$ on average), con-

*Table 3.* Performance on the dev set of GrailQA. The **Bold** and underlined numbers indicate the best and second-best performance.

| Method | LLM | I.I.D | | Compositional | | Zero-shot | | Overall | |
|---|---|---|---|---|---|---|---|---|---|
| | | EM | F1 | EM | F1 | EM | F1 | EM | F1 |
| *Prompting Methods* | | | | | | | | | |
| KB-BINDER (Li et al., 2023) | Codex-davinci-002 | 40.0 | 43.3 | 33.9 | 36.6 | 40.1 | 44.0 | 38.7 | 42.2 |
| KB-Coder (Nie et al., 2024) | GPT-3.5-turbo | 40.6 | 45.5 | 34.5 | 38.6 | 42.2 | 47.3 | 40.1 | 44.9 |
| ARG-KBQA (Tian et al., 2024) | GPT-3.5-turbo | 46.6 | 51.5 | 36.4 | 41.8 | 46.6 | 52.1 | 43.8 | 48.5 |
| *Fine-tune-based Methods* | | | | | | | | | |
| RnG-KBQA (Ye et al., 2022) | T5-large | 86.7 | 89.0 | 61.7 | 68.9 | 68.8 | 74.7 | 69.5 | 76.9 |
| DecAF (Yu et al., 2022) | T5-large | 88.7 | **92.4** | 71.5 | 79.8 | 65.9 | 77.3 | 72.5 | 81.4 |
| TIARA (Shu et al., 2022) | T5-large | 88.4 | 91.2 | 66.4 | 74.8 | 73.3 | 80.7 | 75.3 | 81.9 |
| KBQA-o1 (Luo et al., 2025c) | Llama-3.1-8B | 77.8 $\pm$0.5 | 85.5 $\pm$0.4 | 76.3 $\pm$0.6 | 77.6 $\pm$0.5 | 68.1 $\pm$0.8 | 76.1 $\pm$0.4 | 71.9 $\pm$0.3 | 78.5 $\pm$1.0 |
| **KBQA-R1** | Llama-3.1-8B | **90.0** $\pm$0.6 | 91.5 $\pm$0.4 | **78.0** $\pm$0.4 | **82.5** $\pm$0.7 | **83.6** $\pm$0.3 | **85.2** $\pm$0.4 | **83.9** $\pm$0.4 | **86.1** $\pm$0.6 |
| *Improv. over KBQA-o1* | | +12.8% | +7.0% | +1.7% | +6.3% | +15.5% | +9.1% | +12.0% | +7.6% |

*Table 4.* Results on the test set of WebQSP. The **Bold** and underlined numbers indicate the best and second-best performance.

| Method | LLM | F1 |
|---|---|---|
| *Prompting Methods* | | |
| KB-BINDER (Li et al., 2023) | Codex-davinci-002 | 52.6 |
| KB-Coder (Nie et al., 2024) | GPT-3.5-turbo | 55.7 |
| ARG-KBQA (Tian et al., 2024) | GPT-3.5-turbo | 58.8 |
| Interactive-KBQA (Xiong et al., 2024) | GPT-4-turbo | 71.2 |
| *Graph Retrieval / Reasoning Methods* | | |
| RoG (Luo et al., 2023) | Llama2-7B | 70.8 |
| GNN-RAG (Mavromatis & Karypis, 2025) | Llama2-7B | 71.3 |
| SubgraphRAG (Li et al., 2024b) | Llama-3.1-8B | 70.6 |
| SubgraphRAG (Li et al., 2024b) | GPT-4o | 78.2 |
| *Fine-tune-based Methods* | | |
| RnG-KBQA (Ye et al., 2022) | T5-large | 75.6 |
| DecAF (Yu et al., 2022) | T5-large | 76.7 |
| TIARA (Shu et al., 2022) | T5-large | 78.9 |
| MCTS-KBQA (Gu et al., 2023a) | Llama-3.1-8B | 76.0 |
| KBQA-o1 (Luo et al., 2025c) | Llama-3.1-8B | 57.8 |
| **KBQA-R1** | Llama-3.1-8B | **83.4** $\pm$0.3 |
| *Improv. over KBQA-o1* | | +25.6% |

*Table 5.* Results on the test set of GraphQ. The **Bold** and underlined numbers indicate the best and second-best performance.

| Method | LLM | F1 |
|---|---|---|
| *Prompting Methods* | | |
| KB-BINDER (Li et al., 2023) | Codex-davinci-002 | 27.1 |
| KB-Coder (Nie et al., 2024) | GPT-3.5-turbo | 31.1 |
| *Fine-tune-based Methods* | | |
| SPARQA (Sun et al., 2020) | BERT-base | 21.5 |
| BERT+Ranking (Gu et al., 2021) | BERT-base | 25.0 |
| ArcaneQA (Gu & Su, 2022) | BERT-base | 31.8 |
| CoTKR (Wu et al., 2024) | Llama-3-8B | 47.5 |
| KBQA-o1 (Luo et al., 2025c) | Llama-3.1-8B | 48.7 |
| **KBQA-R1** | Llama-3.1-8B | **53.8** $\pm$0.7 |
| *Improv. over KBQA-o1* | | +5.1% |

firming that iterative refinement through KB feedback is essential. Single-turn generation forces the model to produce complete S-Expressions without intermediate validation, resulting in cascading errors. GrailQA shows the steepest drop ($-36.3\%$), as its compositional and zero-shot questions inherently require exploratory reasoning that cannot be captured in a single generation step.

**Training Strategy Ablations.** Both training components contribute meaningfully to final performance. (1) *w/o RRS* (using standard rejection sampling instead of Referenced Rejection Sampling) reduces average F1 by about 5.6%. This validates our hypothesis that leveraging reference action list during warm-start trajectory generation produces higher-quality training signals. Standard rejection sampling often generates syntactically valid but semantically suboptimal trajectories that provide weaker supervision. (2) *w/o SFT warm-start* (training RL from scratch) incurs a larger penalty (about $-8.6\%$ on average). Without warm-start initialization, the RL agent begins with near-random behavior, requiring substantially more exploration to discover viable reasoning strategies.

**Reward Design Ablations.** Removing the format reward ($r_{\text{format}} = 0$) causes a moderate but consistent drop (about $-2.1\%$ on average). The format reward supplies dense intermediate feedback that steers the agent toward syntactically well-formed actions and encourages necessary thinking before acting, thereby complementing the sparse outcome reward. Without this signal, the agent can produce incorrect tag ordering or incomplete tags, which prevent the system from correctly extracting information. The relatively smaller impact compared to architectural ablations suggests that the outcome reward remains the primary driver of learning, with format rewards serving as a stabilizing auxiliary signal.

**Referenced RS vs. Standard RS** To better understand the effect of Referenced Rejection Sampling (RRS) compared to standard Rejection Sampling (RS), we compare three aspects of the training pipeline on all three datasets: (1) the raw F1 score obtained directly from RS trajectories before any SFT, (2) the number of trajectories that pass both the outcome filter (F1 > 0.9) and the structure reward filter ($r_{\text{format}} = 0.1$) and are used for SFT, and (3) the initial test-set F1 after SFT trained on the corresponding RS data. As

*Table 6.* Component ablation study of KBQA-R1. We report Overall F1 (%) on three datasets.

| Variant | WebQSP | GraphQ | GrailQA |
|---|---|---|---|
| Full KBQA-R1 (ours) | **83.4** | **53.8** | **86.1** |
| *Agent Architecture Ablations* | | | |
| w/o RRCG (no relation retrieval & gating) | 64.1 | 37.7 | 67.1 |
| w/o Multi-turn (single-turn action generation) | 63.2 | 34.1 | 49.8 |
| *Training Strategy Ablations* | | | |
| w/o RRS (standard rejection sampling) | 78.9 | 49.2 | 78.3 |
| w/o SFT warm-start (RL from scratch) | 75.2 | 47.3 | 75.1 |
| w/o GRPO (only SFT) | 72.1 | 47.8 | 80.2 |
| *Reward Design Ablations* | | | |
| w/o Format Reward ($r_{format} = 0$) | 81.1 | 51.6 | 84.2 |

*Table 7.* Standard Rejection Sampling (RS) vs. Referenced RS (RRS) across three datasets. "RS F1 (pre-SFT)" is the average F1 of raw RS trajectories before fine-tuning. "Filtered SFT Samples" counts trajectories with F1 > 0.9 and $r_{format} = 0.1$ used for SFT. "SFT Init F1" reports dev-set F1 after SFT initialized from the corresponding RS data.

| Dataset | Method | RS F1 (pre-SFT) | # Accepted / Total | Acceptance (%) | SFT Init F1 |
|---|---|---|---|---|---|
| GrailQA | Standard RS | 54.2 | 17248 / 43851 | 39.3 | 73.8 |
| | Referenced RS (RRS) | 70.2 | 29384 / 43851 | 67.0 | 80.2 |
| WebQSP | Standard RS | 49.1 | 1120 / 2929 | 38.3 | 65.8 |
| | Referenced RS (RRS) | 62.5 | 1505 / 2929 | 51.4 | 72.1 |
| GraphQ | Standard RS | 48.1 | 986 / 2332 | 42.3 | 41.1 |
| | Referenced RS (RRS) | 73.1 | 1562 / 2332 | 67.0 | 47.8 |

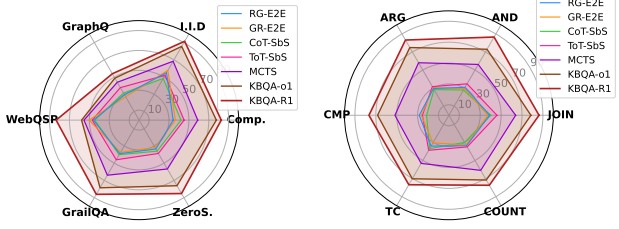

*(a)* F1 scores across datasets and generalization levels. *(b)* F1 scores across logical operation types.

*Figure 3.* Comprehensive performance comparison of KBQA-R1 with baseline methods using Llama-3.1-8B.

shown in Table 7, RRS consistently improves the quality and efficiency of trajectory collection across all datasets. The acceptance statistics reveal that RRS yields markedly more usable trajectories under the same filtering criteria, demonstrating that RRS is substantially more sample-efficient than standard RS. Finally, these higher-quality and denser trajectories translate into stronger SFT initialization. Starting RL from an RRS-initialized SFT checkpoint places the policy closer to a good solution, which complements the ablation result that removing RRS leads to a noticeable drop in final performance. Together, these observations justify RRS as a key component for obtaining stable and high-performing RL training in KBQA-R1.

## 5.4. Compared with Llama-3.1-8B based Methods

Following the experimental setup of KBQA-o1 (Luo et al., 2025c), we conduct a focused comparison among methods that share the same Llama-3.1-8B backbone and Freebase execution environment. The compared baselines can be grouped into three categories. (1) *End-to-end generation methods*: RG-E2E and GR-E2E are adapted from DecAF (Yu et al., 2022) and ChatKBQA (Luo et al., 2024), respectively. RG-E2E follows a retrieve-then-generate paradigm, while GR-E2E first generates a preliminary logical form and then refines it with KB retrieval. (2) *Step-by-step prompting methods*: CoT-SbS and ToT-SbS are implemented by instantiating the CoT-based QueryAgent (Huang et al., 2024) and the ToT-based ToG framework (Sun et al., 2023) on Llama-3.1-8B, prompting the model to alternate between intermediate thoughts and KB queries. (3) *MCTS-based agentic method*: MCTS corresponds to the MCTS-optimized variant in KBQA-o1 (Luo et al., 2025c) without incremental Finetuning. Figure 3a visualizes F1 scores across six evaluation dimensions. KBQA-R1 achieves the largest coverage area, demonstrating superior overall performance across all settings, with the most pronounced gap in zero-shot dimensions. This validates our hypothesis that RL-based training fosters more robust reasoning capabilities than SFT. In contrast, end-to-end and step-by-step baselines cluster in the inner region, reflecting limited generalization. Figure 3b breaks down performance by logical operation type. KBQA-R1 dominates across all categories, showing significant advantages in complex operations. Conversely,

*Table 8.* Comparison with frontier generalist agents on the 500-query subset. "Harness" uses the KBQA-R1 action space and feedback format with an off-the-shelf model, while KBQA-R1 uses the learned RL policy.

| Group | Model | WebQSP | GrailQA | GraphQ | Avg. | Turns | Tokens In / Out |
|---|---|---|---|---|---|---|---|
| SPARQL ReAct | GLM-5 | 48.6 | 63.8 | 37.9 | 52.5 | 7.10 | 10,792 / 1,952 |
| SPARQL ReAct | Kimi-K2.5 | 49.1 | 62.3 | 36.5 | 51.9 | 6.05 | 7,583 / 4,615 |
| SPARQL ReAct | gpt-5.3-codex | 54.5 | 65.6 | 42.7 | 56.6 | 5.20 | 4,994 / 454 |
| SPARQL ReAct | Claude 4.6 Sonnet | 54.2 | 65.4 | 36.4 | 55.1 | 5.79 | 8,546 / 824 |
| KBQA-R1 Harness | GLM-5 | 64.5 | 70.2 | 48.5 | 63.6 | 8.43 | 14,235 / 977 |
| KBQA-R1 Harness | Kimi-K2.5 | 73.3 | 73.9 | 53.3 | 69.5 | 7.24 | 12,880 / 2,928 |
| KBQA-R1 Harness | gpt-5.3-codex | 71.1 | 73.5 | 51.5 | 68.1 | 3.43 | 4,537 / 374 |
| KBQA-R1 Harness | Claude 4.6 Sonnet | 76.7 | 76.3 | 51.8 | 71.6 | 5.45 | 15,325 / 2,959 |
| KBQA-R1 | Llama-3.1-8B | **84.2** | **85.6** | **55.4** | **79.0** | **2.91** | **3,994 / 328** |

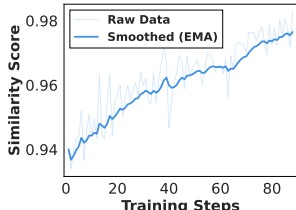

*(a)* Relation similarity score evolution during training.

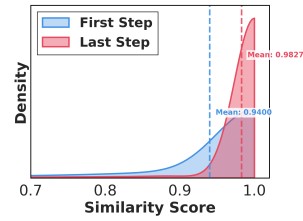

*(b)* Distribution shift of similarity scores: first step vs. last step.

*Figure 4.* Relation similarity analysis during RL.

baselines struggle with rare operations, underscoring their inability to generalize to infrequent query patterns.

**Frontier-agent comparison.** We further evaluate whether a strong off-the-shelf agent can close the gap when equipped with the same KBQA-R1 harness. The experiments in Table 8 use a fixed 500-query subset (200 GrailQA, 200 WebQSP, and 100 GraphQ) because multi-turn proprietary-agent evaluation is expensive. The structured harness improves every frontier model over direct SPARQL ReAct, confirming that the action space and feedback state are useful independently of RL. However, even the best harness-equipped frontier model remains below the learned KBQA-R1 policy while using more turns and substantially more tokens, indicating that KBQA-R1 internalizes graph navigation instead of relying on test-time trial and error.

### 5.5. Training Dynamics Analysis

Figure 4a shows that the top-1 relation similarity of WebQSP dataset improves steadily from ∼0.94 at the SFT warm-start checkpoint to ∼0.98 at convergence, indicating that GRPO consistently strengthens relation grounding rather than introducing unstable fluctuations. Figure 4b further confirms this trend: compared with the first step (mean=0.9400), the last-step distribution becomes more concentrated near 1.0 (mean=0.9827) with reduced variance, suggesting fewer low-confidence relation choices and more

reliable query generation. We defer the training reward curve and its detailed discussion to Appendix B.6.

## 6. Conclusion

We presented KBQA-R1, a reinforcement learning framework for agentic knowledge base question answering. By integrating a structured action space, a relation retrieval and confidence gating module, and a novel Referenced Rejection Sampling warm-start strategy, KBQA-R1 effectively leverages execution feedback from the knowledge base to learn robust reasoning policies via the GRPO algorithm. Extensive experiments on three challenging KBQA benchmarks demonstrate that KBQA-R1 significantly outperforms state-of-the-art prompting and fine-tuning baselines, particularly in out-of-distribution generalization settings.

## Impact Statement

This work aims to improve reliable question answering over structured knowledge bases by training LLM agents to ground decisions in executable KB actions and execution feedback. Its main positive impact is more accurate and auditable access to factual knowledge, with lower test-time cost than search-heavy agentic systems. Such systems can support scientific, enterprise, and educational knowledge access when the underlying KB is licensed, maintained, and appropriately scoped.

Potential risks mainly arise from deployment context rather than from the benchmark experiments themselves. KBQA-R1 can still return incomplete or misleading answers when entity linking fails, when the KB lacks relevant facts, or when the graph contains outdated or biased information. Because the method shifts part of the search burden from inference to RL training with online KB execution, practitioners should also consider the energy and infrastructure cost of retraining on new domains. All experiments use public benchmarks and public knowledge bases, but downstream

deployments should monitor answer provenance, surface uncertainty or execution failures, respect KB licensing and privacy constraints, and avoid using KBQA outputs as the sole basis for high-stakes decisions.

## Acknowledgements

This work was supported by the Beijing Natural Science Foundation (L252033) and the National Natural Science Foundation of China (62576339, 92570204).

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

# A. Extended Related Work

### A.1. Knowledge Base Question Answering (KBQA).

Before the rise of LLMs, KBQA studies are commonly categorized into information-retrieval-based (IR-based) methods (Sun et al., 2018; 2019; Zhang et al., 2022; He et al., 2021; Saxena et al., 2020) and semantic-parsing-based (SP-based) methods (Ye et al., 2022; Shu et al., 2022; Gu & Su, 2022; Zhang et al., 2023a). With LLMs, three paradigms have emerged: (i) *end-to-end approaches* that directly generate logical forms via in-context learning or fine-tuning (Li et al., 2023; Nie et al., 2024; Luo et al., 2024; Jiang et al., 2023); (ii) *step-by-step (agentic) approaches* that interleave reasoning with graph exploration and tool use (Gu et al., 2023b; Huang et al., 2024; Sun et al., 2023; Luo et al., 2023; Xiong et al., 2024; Chen et al., 2024; Jiang et al., 2024); and (iii) *search-augmented approaches* that leverage tree search algorithms such as Monte Carlo Tree Search (MCTS) for systematic exploration (Luo et al., 2025c).

While MCTS-based methods like KBQA-o1 (Luo et al., 2025c) achieve strong performance through heuristic exploration, they exhibit two key limitations. **First, they incur significant computational overhead from multiple rollouts per query and require separate policy and reward models during inference. Second, their reasoning traces are often *template-driven* (e.g., "At this step, we should find the relation...") rather than genuinely analytical—the model announces *what* action to take without explaining *why* based on observations. In contrast, we train a single policy via RL with outcome-based rewards, encouraging the model to develop *adaptive reasoning* that analyzes environmental feedback and justifies action choices, while eliminating test-time search overhead.**

### A.2. LLMs, tool use, and agentic reasoning.

Chain-of-Thought (CoT) prompting improves reasoning by eliciting intermediate steps (Wei et al., 2022); ReAct (Yao et al., 2022) interleaves "think" and "act" to ground reasoning in environment feedback; and heuristic search has been applied to agent traces (e.g., MCTS-style selection in (Hao et al., 2023) and tree-structured deliberation in (Yao et al., 2023)). Recent graph-augmented approaches such as Plan-on-Graph (Chen et al., 2024) incorporate self-correcting mechanisms with dynamic memory for adaptive planning on knowledge graphs. While these methods expand the search space or stabilize multi-step reasoning, free-form thoughts can overfit prompt templates and do not guarantee executability. We keep the interleaved think-act design but require typed, schema-aware actions with validators and an executor, turning traces into verifiable computations rather than narrative justifications.

### A.3. Retrieval-augmented generation and search-as-a-tool.

Classical RAG pipelines retrieve text snippets and feed them to the model for generation (Lewis et al., 2020). Recent work moves toward search-as-a-tool, prompting or training LLMs to issue search calls and iterate (Trivedi et al., 2022; Yao et al., 2022; Schick et al., 2023; Li et al., 2025; Jin et al., 2025; Luo et al., 2025a). GraphRAG and graph-decoding approaches (He et al., 2024; Ma et al., 2025; Mavromatis & Karypis, 2025; Li et al., 2024b;a; Luo et al., 2025a;b) further integrate graph retrieval with LLM reasoning, enabling tighter coupling between structured knowledge and text-based evidence. These approaches reduce hallucination but depend heavily on retrieval quality, local subgraph construction, or constrained decoding at inference time. Our setting differs fundamentally by treating a *knowledge graph* as the environment: actions are typed and executable against the KB schema, observations are structure-grounded entity sets rather than text passages, and step-wise executability can be validated programmatically rather than inferred from unstructured documents.

# B. Experimental Details

### B.1. Datasets and Statistics

We conduct experiments on three widely-used KBQA benchmarks, each designed to evaluate different aspects of model generalization and reasoning capabilities. All datasets are grounded on Freebase (Bollacker et al., 2008). **GrailQA** (Gu et al., 2021) is a large-scale dataset specifically designed to evaluate KBQA models across three generalization levels: *i.i.d.*, *compositional*, and *zero-shot*. It contains 64,331 questions in total, with 44,337 training questions, 13,231 validation questions and 6,763 test questions. Following prior work (Luo et al., 2024; 2025c), we use the dev set for evaluation. The compositional and zero-shot settings are particularly challenging, requiring models to handle unseen combinations of entities and relations. **WebQSP** (Yih et al., 2016) is an enriched version of WebQuestions, providing semantic parses for 4,737 questions. The dataset is split into 3,098 training questions and 1,639 test questions. **GraphQuestions** (Su et al., 2016) tests

KBQA models on complex graph-structured reasoning. It contains 5,166 questions in total, with 2,508 for training and 2,658 for testing. The dataset challenges models to navigate multi-hop relationships.

### B.2. Baselines (Detailed Descriptions)

We compare KBQA-R1 with both *fine-tune-based* and *prompting-based* KBQA methods.

**Fine-tune-based methods.** These methods are trained on the full supervision of the corresponding benchmark and are included as upper-bound references.

- **RnG-KBQA** (Ye et al., 2022): a retrieve-and-generate framework that first retrieves relevant KB evidence and then generates executable logical forms.

- **DecAF** (Yu et al., 2022): a multi-granular retrieval and refinement approach that progressively improves retrieved knowledge for robust KBQA.

- **TIARA** (Shu et al., 2022): a semantic parsing method that maps questions to structured queries through iterative refinement.

- **CoTKR** (Wu et al., 2024): a Chain-of-Thought enhanced knowledge rewriting method that trains a knowledge rewriter via SFT and DPO to convert KB triples into natural language reasoning text for improved QA.

- **KBQA-o1** (Luo et al., 2025c): an agentic KBQA approach based on Monte-Carlo Tree Search (MCTS) with policy and reward models for heuristic search.

For GraphQuestions, following the setup in KBQA-o1 (Luo et al., 2025c), we additionally report results of **SPARQA** (Sun et al., 2020), **BERT+Ranking** (Gu et al., 2021), and **ArcaneQA** (Gu & Su, 2022).

**Prompting-based methods.** These methods operate under limited or no task-specific gradient updates and are compared under a similar low-annotation constraint.

- **KB-BINDER** (Li et al., 2023): an in-context learning method (GPT-3.5-turbo) that binds questions to KB entities and relations.

- **KB-Coder** (Nie et al., 2024): a code-style in-context learning method (GPT-3.5-turbo) that generates executable logical forms.

- **ARG-KBQA** (Tian et al., 2024): a prompting approach (GPT-3.5-turbo) that uses augmented reasoning graphs to improve KBQA.

- **Interactive-KBQA** (Xiong et al., 2024): an interactive framework that decomposes complex questions into sub-questions and iteratively queries the KB to gather evidence for answer generation.

- **ToG (Think-on-Graph)** (Sun et al., 2023): a beam search-based method that prompts LLMs to explore multiple reasoning paths on the knowledge graph, selecting the most promising path at each step.

- **PoG (Plan-on-Graph)** (Chen et al., 2024): an advanced graph reasoning method that first generates a reasoning plan and then executes it on the knowledge graph with self-correction capabilities.

- **RoG** (Luo et al., 2023): a reasoning-on-graphs framework that asks LLMs to reason over graph-derived paths and evidence.

- **GNN-RAG** (Mavromatis & Karypis, 2025): a graph neural retrieval method that retrieves question-relevant graph paths before LLM reasoning.

- **SubgraphRAG** (Li et al., 2024b): a graph-centric RAG method that retrieves compact subgraphs for LLM answer prediction.

- **DoG** (Li et al., 2024a): a constrained graph-decoding method that encourages LLMs to generate well-formed chains on knowledge graphs.

## B.3. Training Hyperparameters and Infrastructure

### B.3.1. GRPO CONFIGURATION

We adopt the GRPO algorithm (Shao et al., 2024) with the following hyperparameters: (1) *Rollout sampling*: $n = 5$ responses per prompt with temperature $\tau = 1.0$ and top-$p = 1.0$; (2) *Clipping*: asymmetric clip ratios $\epsilon_{\text{low}} = 0.2$, $\epsilon_{\text{high}} = 0.28$ following DAPO (Yu et al., 2025); (3) *KL regularization*: KL loss coefficient $\beta = 0.001$; (4) *Reward weights and gating*: we set $\lambda_{\text{outcome}} = 1.0$ and $\lambda_{\text{format}} = 0.1$, and use RRCG confidence thresholds $\tau_{\text{high}} = 0.95$ and $\tau_{\text{low}} = 0.3$ across all datasets; (5) *Batch configuration*: train batch size 256, PPO mini-batch size 128, with dynamic micro-batching enabled.

### B.3.2. INFRASTRUCTURE

Training is conducted on $8\times$NVIDIA A100-80GB GPUs with FSDP (Zhao et al., 2023) for model sharding. The Freebase KB backend uses Virtuoso (Erling & Mikhailov, 2009) with ODBC connection pooling (pool size 48, query timeout 600s).

*Table 9.* Hyperparameters for KBQA-R1 training and inference.

| Category | Hyperparameter | Value |
|---|---|---|
| GRPO | Rollout samples per prompt ($n$) | 5 |
| | Temperature ($\tau$) | 1.0 |
| | Top-$p$ | 1.0 |
| | Clip ratio (low / high) | 0.2 / 0.28 |
| | KL coefficient ($\beta$) | 0.001 |
| Reward | Outcome weight ($\lambda_{\text{outcome}}$) | 1.0 |
| | Format weight ($\lambda_{\text{format}}$) | 0.1 |
| RRCG | High threshold ($\tau_{\text{high}}$) | 0.95 |
| | Low threshold ($\tau_{\text{low}}$) | 0.3 |
| Batch | Train batch size | 256 |
| | PPO mini-batch size | 128 |
| Inference | Max prompt length | 14,336 |
| | Max response length | 1,024 |
| | Max agent turns | 6 |

*Table 10.* Average number of LLM forward calls per question. ToG and PoG results are from (Chen et al., 2024).

| Dataset | Method | Avg. LLM calls ↓ |
|---|---|---|
| WebQSP | ToG (GPT-4) | 15.9 |
| | PoG (GPT-4) | 9.0 |
| | KBQA-o1 (Llama-3.1-8B) | 28.8 |
| | KBQA-R1 (Llama-3.1-8B) | **2.65** |
| GrailQA | ToG (GPT-4) | 11.1 |
| | PoG (GPT-4) | 6.5 |
| | KBQA-o1 (Llama-3.1-8B) | 32.3 |
| | KBQA-R1 (Llama-3.1-8B) | **3.08** |
| GraphQ | KBQA-o1 (Llama-3.1-8B) | 78.0 |
| | KBQA-R1 (Llama-3.1-8B) | **3.16** |

## B.4. LLM Call Efficiency

To quantify the computational overhead between KBQA-R1 and existing methods, we compare the number of LLM forward calls required during inference. Table 10 reports average calls per question on 200 randomly sampled examples for each dataset.

**Comparison with MCTS-based Methods.** KBQA-o1 performs many LLM calls per query and additionally invokes separate policy and reward models, leading to substantially more LLM evaluations. In contrast, KBQA-R1 uses a single GRPO-trained policy without test-time search, reducing LLM calls by over 80% while achieving higher accuracy. **In a 8-A100 GPU setup, KBQA-R1 processes about 155.6 questions per minute on GrailQA, compared to only 5.9 questions per minute for KBQA-o1.**

**Comparison with GPT-4 Prompting Methods.** We further compare with state-of-the-art prompting-based methods ToG (Sun et al., 2023) and PoG (Chen et al., 2024) that leverage GPT-4 for knowledge graph reasoning. As shown in

*Table 11.* Hits@1 (%) comparison with graph reasoning methods on GrailQA and WebQSP. ToG and PoG results are from (Chen et al., 2024).

| Method | LLM | WebQSP | GrailQA | | | |
|---|---|---|---|---|---|---|
| | | | Overall | I.I.D | Comp. | Zero-shot |
| *Prompting KG-Augmented LLM w/GPT-3.5* | | | | | | |
| ToG (Sun et al., 2023) | GPT-3.5 | 76.2 | 68.7 | 70.1 | 56.1 | 72.7 |
| PoG (Chen et al., 2024) | GPT-3.5 | 82.0 | 76.5 | 76.3 | 62.1 | 81.7 |
| *Prompting KG-Augmented LLM w/GPT-4* | | | | | | |
| ToG (Sun et al., 2023) | GPT-4 | 82.6 | 81.4 | 79.4 | 67.3 | 86.5 |
| PoG (Chen et al., 2024) | GPT-4 | 87.3 | 84.7 | 87.9 | 69.7 | 88.6 |
| *Fine-tuned Methods* | | | | | | |
| KBQA-R1 (Ours) | Llama-3.1-8B | 88.2 | 86.2 | 91.2 | 80.1 | 86.7 |

*Figure 5.* Training reward curve on GrailQA during GRPO.

Table 10, ToG requires 15.9 and 11.1 LLM calls per sample on WebQSP and GrailQA respectively, while PoG requires 9.0 and 6.5 calls. In contrast, KBQA-R1 achieves **2.65** and **3.08** calls on the same datasets—a reduction of over 50% compared to PoG and over 70% compared to ToG. This significant efficiency gain stems from RL training, which enables the model to learn **precise, goal-directed navigation** on the knowledge graph rather than relying on exploratory search or iterative self-correction. The GRPO-trained policy internalizes effective reasoning strategies, allowing it to reach correct answers with fewer interaction steps.

### B.5. Extended System Comparisons

We conduct supplementary evaluations to clarify the role of RL training and backend transfer. Unless otherwise noted, scores are F1 percentages.

*Table 12.* Comparison with a Search-R1-style RL baseline adapted to the same action space and executor.

| Method | Warm-start / Gating | WebQSP | GraphQ | GrailQA |
|---|---|---|---|---|
| Search-R1 adapted | Standard RS, no RRCG | 57.0 | 30.3 | 54.1 |
| KBQA-R1 | RRS + RRCG | **83.4** | **53.8** | **86.1** |

*Table 13.* WebQSP Hits@1 comparison after transferring the executor backend from Freebase to Wikidata.

| Method | LLM | KG Backend | Hits@1 |
|---|---|---|---|
| ToG | GPT-3.5 | Freebase | 76.2 |
| ToG | GPT-3.5 | Wikidata | 68.6 |
| KBQA-R1 | Llama-3.1-8B | Freebase | 88.2 |
| KBQA-R1 | Llama-3.1-8B | Wikidata | **77.3** |

---

**Algorithm 1** RRS Warm-Start + GRPO Training Recipe

---

**Require:** Base LLM $\pi_{\text{base}}$, Training data $D = \{(q_i, \mathcal{A}_i, S_i^*)\}$, Executor $\mathcal{E}$, Reward threshold $\tau$, Num rollouts $n$
**Ensure:** Optimized policy $\pi_\theta$
 1: **// Phase 1: Referenced Rejection Sampling (RRS)**
 2: $S_{\text{RRS}} \leftarrow \emptyset$
 3: **for** each $(q, \mathcal{A}, S^*)$ in $D$ **do**
 4:     Extract ground-truth action sequence $\mathbf{a}^* = (a_1^*, \ldots, a_k^*)$ from $S^*$
 5:     *// Run referenced rollout with ground-truth actions as guidance*
 6:     Generate trajectory $y$ conditioned on $\mathbf{a}^*$ using $\mathcal{E}$
 7:     **if** $R(y) \geq \tau$ and valid S-Expression **then**
 8:         Strip reference hints from $y$; add to $S_{\text{RRS}}$
 9:     **end if**
10: **end for**
11: **// Phase 2: SFT Warm-Start**
12: $\pi_{\theta_0} \leftarrow \text{SFT}(\pi_{\text{base}}, S_{\text{RRS}})$
13: **// Phase 3: Reinforcement Learning (GRPO)**
14: Initialize policy $\pi_\theta \leftarrow \pi_{\theta_0}$
15: Initialize reference policy $\pi_{\text{ref}} \leftarrow \pi_{\theta_0}$
16: **for** RL iteration $k = 1, \ldots, N$ **do**
17:     Sample batch of prompts $\{x\}$ from $D$
18:     **for** each prompt $x$ in batch **do**
19:         Generate $n$ trajectories $\{y_i\}_{i=1}^n \sim \pi_\theta(\cdot|x)$ using $\mathcal{E}$
20:         Compute rewards $\{R_i\}_{i=1}^n$ using composite reward $R$
21:         Compute advantages $\{\hat{A}_i\}_{i=1}^n$ where $\hat{A}_i = R_i - \frac{1}{n}\sum_{j=1}^n R_j$
22:     **end for**
23:     Update $\pi_\theta$ by maximizing: $J_{\text{GRPO}} = \mathbb{E}[\min(r_t \hat{A}_t, \text{clip}(r_t, 1-\epsilon, 1+\epsilon)\hat{A}_t)] - \beta D_{\text{KL}}$
24: **end for**
25: **return** $\pi_\theta$

---

## B.6. Training Dynamics

Figure 5 reports the training reward curve during GRPO on GrailQA. The rewards mean curves shows a clear upward trend as training progresses, and stabilizes toward the end of training, indicating policy convergence. The reward signal, which combines outcome reward (F1-based) and format reward, shows a clear upward trajectory from approximately 0.89 to 1.00. The reward briefly decreases during steps 0-5, reflecting the exploration phase where the agent deviates from the SFT-initialized policy to discover potentially better strategies. Between steps 5-60, the reward increases rapidly, indicating successful policy refinement through the GRPO objective. After step 140, the reward stabilizes around 1.00 with reduced variance. Given the maximum achievable reward of 1.10 (1.0 for outcome and 0.1 for structure), this suggests that the policy has converged to a near-optimal state.

## B.7. Comparison with SOTA Graph Reasoning Methods (Hits@1)

We additionally compare KBQA-R1 with prompting-based graph reasoning methods ToG (Sun et al., 2023) and PoG (Chen et al., 2024) using the Hits@1 metric, which measures whether the top-1 predicted answer matches the gold answer. As shown in Table 11, we report results on GrailQA and WebQSP following the experimental setup in (Chen et al., 2024). Note that ToG and PoG use Hits@1 as their primary metric, which differs from the F1 metric used in our main experiments. As

---

**Algorithm 2** KBQA-R1 Multi-Turn Rollout

---

**Require:** Question $q$, policy model $\pi_\theta$, executor $\mathcal{E}$, retrieval $\mathcal{R}$, thresholds $(\tau_{\text{low}}, \tau_{\text{high}})$, max turns $T$
**Ensure:** Answer, trajectory
 1: Initialize context $C_0$ from template$(q)$; function list $F \leftarrow \emptyset$; $t \leftarrow 0$
 2: **while** $t < T$ **do**
 3:     Generate continuation $y_t$ with $\pi_\theta$ until termination token
 4:     **if** `</answer>` in $y_t$ **then**
 5:         Extract $\hat{\mathcal{A}}$; **break**
 6:     **end if**
 7:     Parse actions $A_t = \{a^{(1)}, \ldots, a^{(m)}\}$ from `<action>`
 8:     **for** $i = 1$ to $m$ **do**
 9:         **if** action contains relation parameter **then**
10:             Retrieve via $\mathcal{R}$; let $r_{\text{best}}, s_{\text{best}}$ be top-1
11:             **if** $s_{\text{best}} < \tau_{\text{low}}$ **then**
12:                 Mark invalid; **continue** {Rejected}
13:             **else if** $s_{\text{best}} < \tau_{\text{high}}$ **then**
14:                 Flag uncertainty {Tentative}
15:             **end if**
16:         **end if**
17:         **if** action is `Find_relation` **then**
18:             Append JOIN$(r_{\text{best}}, \text{expr/entity})$ to $F$
19:         **else if** action is `Merge` **then**
20:             Append AND$(\text{expr}_1, \text{expr}_2)$ to $F$
21:         **else if** action is `Compare` **then**
22:             Append CMP$(mode, r_{\text{best}}, \text{expr})$ to $F$
23:         **else if** action is `Time_constraint` **then**
24:             Append TC$(\text{expr}, r_{\text{best}}, time)$ to $F$
25:         **else if** action is `Order` **then**
26:             Append ARG$(mode, \text{expr}, r_{\text{best}})$ to $F$
27:         **else if** action is `Count` **then**
28:             Append COUNT$(\text{expr})$ to $F$
29:         **end if**
30:     **end for**
31:     Build S-Expression $S_t$ from $F$; execute with $\mathcal{E}$
32:     Build `<information>` block with results
33:     Append to context $C_{t+1}$; $t \leftarrow t + 1$
34: **end while**
35: **return** answer, trajectory

---

shown in Table 11, KBQA-R1 achieves competitive or superior performance compared to GPT-4-based methods despite using a significantly smaller Llama-3.1-8B backbone. On WebQSP, KBQA-R1 achieves 88.2% Hits@1, outperforming PoG (GPT-4) by 0.9%. On GrailQA, KBQA-R1 demonstrates particularly strong performance on the I.I.D (91.2%) and Compositional (80.1%) settings, surpassing PoG (GPT-4) by 3.3% and 10.4% respectively. These results highlight the effectiveness of RL-based training: by learning precise knowledge graph navigation through environmental feedback, KBQA-R1 can match or exceed the reasoning capabilities of much larger commercial models while requiring substantially fewer LLM calls (see Table 10).

### B.8. Algorithm Pseudocode

We provide detailed pseudocode for two core components of KBQA-R1. Algorithm 1 presents the complete training recipe, which includes three phases: (1) Referenced Rejection Sampling (RRS) for generating high-quality warm-start data, (2) SFT warm-start to initialize the policy, and (3) GRPO-based reinforcement learning for policy optimization. Algorithm 2 describes the multi-turn rollout procedure during inference, detailing how the agent interacts with the knowledge base

through action parsing, relation retrieval with confidence gating (RRCG), and S-Expression construction.

## B.9. Sample Interaction Trace

We provide a complete example of a KBQA-R1 agent interaction trace to illustrate the multi-turn reasoning process. The question is drawn from GrailQA. This example showcases four interaction turns, including an unsuccessful action attempt and subsequent error recovery, demonstrating the agent's ability to adapt its strategy based on environment feedback. We highlight four key modules: `<think>` (model reasoning), `<action>` (KB queries), `<information>` (environment feedback), and `<answer>` (final output). The agent achieves F1 = 1.0 on this example.

---

**PROMPT**

You are an expert assistant for querying the Freebase knowledge base using structured reasoning actions. Answer the given question about Freebase knowledge base. You **must** conduct reasoning inside `<think>...</think>` first every time you get new information. After reasoning, provide structured actions inside `<action>...</action>` tags. The system will return query results between `<information>...</information>`. When ready, provide the answer inside `<answer>...</answer>` using MIDs or literal values.

**Available Actions:** `Find_relation, Merge, Order, Compare, TIME_Constraint, Count` **Topic Entities:** [ 'Doing Clarence a Bit of Good' (`m.09gs5j0`) ] **Question:** What short story has a character who also is in Doing Clarence a Bit of Good?

---

**AGENT — Turn 1**

`<think>` To find the short story that shares a character with "Doing Clarence a Bit of Good," I first need to identify the characters that appear in "Doing Clarence a Bit of Good." `</think>`

`<action>` Find_relation [ m.09gs5j0 | book.book_character.character_appearing ] `</action>`

**ENVIRONMENT FEEDBACK**

```
<information>
functions:
expression1 = START('m.09gs5j0')
expression1 = JOIN('book.book_character.appears_in_stories', expression1)
result_mid_list: [ m.0dzx0p (Reggie Pepper) ]

</information>
```

**AGENT — Turn 2**

`<think>` I found that Reggie Pepper is a character in "Doing Clarence a Bit of Good." Next, I need to find out which short stories this character appears in. `</think>`

`<action>` Find_relation [ m.0dzx0p | book.short_story.characters ] `</action>`

**ENVIRONMENT FEEDBACK**

```
<information>
functions:
expression1 = START('m.09gs5j0')
expression1 = JOIN('book.book_character.appears_in_stories', expression1)
expression2 = START('m.0dzx0p')
expression2 = JOIN('book.short_story.characters', expression2)
result_mid_list: [ m.09gs5hv (Helping Freddie), m.09gs5j0 (Doing Clarence a Bit of Good), m.09gs5hy (Rallying
Round Old George), m.09gs5hr (Absent Treatment), m.010r9gwy (Disentangling Old Duggie), m.010rlht0 (The Test
Case), m.010rlh9d (Concealed Art) ]

</information>
```

**AGENT — Turn 3**

`<think>` I have a list of short stories that feature Reggie Pepper, including "Doing Clarence a Bit of Good." Now, I need to filter this list to only include stories that are of the type "short story" and not "Doing Clarence a Bit of Good" itself. `</think>`

`<action>` Merge [ expression2 | book.short_story ] `</action>`

**ENVIRONMENT FEEDBACK**

```
<information>
functions: ... (previous expressions omitted)
expression3 = START('book.short_story')
expression2 = AND(expression2, expression3)
result_mid_list: [ m.09gs5hv (Helping Freddie), m.09gs5j0 (Doing Clarence a Bit of Good), m.09gs5hy (Rallying
Round Old George), m.09gs5hr (Absent Treatment), m.010r9gwy (Disentangling Old Duggie), m.010rlht0 (The Test
Case), m.010rlh9d (Concealed Art) ]

</information>
```

**AGENT — Turn 4 (Final)**

`<think>` The merged results now confirm all short stories featuring Reggie Pepper. Since the question asks for stories that share a character with "Doing Clarence a Bit of Good," I should exclude the story itself from the final answer and report the remaining six. `</think>`

`<answer>` m.09gs5hv  m.09gs5hy  m.09gs5hr  m.010r9gwy  m.010rlht0  m.010rlh9d `</answer>`

*Figure 6.* Sample KBQA-R1 interaction trace on a GrailQA question. The trace illustrates multi-turn reasoning with executable actions, environment feedback, error recovery, and final answer generation.

## B.10. Case Study: KBQA-o1 vs. KBQA-R1

We compare KBQA-o1 and KBQA-R1 on the *same* GrailQA question to illustrate the impact of grounded environment feedback.

---

**What short story has a character who also is in Doing Clarence a Bit of Good?**

**Topic Entity:** 'Doing Clarence a Bit of Good' (`m.09gs5j0`)  **Gold:** 6 MIDs (`m.09gs5hv, m.09gs5hy, m.09gs5hr, m.010r9gwy, m.010rlht0, m.010rlh9d`)

---

**KBQA-o1 (Incorrect, F1 = 0.0)**

**Step 1**  Thought: *At this step, we should identify a topic entity from the question to start a new expression.* ✗ Rigid template

Action: Extract_entity [ Doing Clarence a Bit of Good ]

Obs: expression = START('m.09gs5j0')                              ✗ Action echo, not KB results

**Step 2**  Thought: *At this step, we should find the one-hop relation connected to the current expression.* ✗ Rigid template

Action: Find_relation [ book.short_story.characters ]

Obs: expression = JOIN('(R book.short_story.characters)', expr)   ✗ Action echo — wrong direction undetected

**Step 3**  Thought: *At this step, we should identify a topic entity from the question.* ✗ Rigid template

Action: Extract_entity [ book.short_story ]

Obs: expression1 = START('book.short_story')                       ✗ Action echo, not KB results

**Step 4**  Thought: *At this step, we conclude that it is appropriate to end and output the expression.* ✗ Rigid template

Action: Finish [ expression1 ]                                     ✗ Finishes expression1, discards JOIN

---

pred_answer = [ ]                                                 ✗ **Empty result (F1 = 0.0)**

---

**KBQA-R1 (Correct, F1 = 1.0)**

**Step 1**  Think: *To find the short story that shares a character with "Doing Clarence a Bit of Good," I first need to identify the characters that appear in "Doing Clarence a Bit of Good."* ✓ Genuine reasoning

Action: Find_relation [ m.09gs5j0 | book.book_character.appears_in_stories ]

Info: result_mid_list: [ m.0dzx0p (Reggie Pepper) ]               ✓ Real KB results returned

**Step 2**  Think: *I found that Reggie Pepper is a character in "Doing Clarence a Bit of Good." Next, I need to find out which short stories this character appears in.* ✓ Adapts based on results

Action: Find_relation [ m.0dzx0p | book.short_story.characters ]

Info: result_mid_list: [ m.09gs5hv (Helping Freddie), m.09gs5j0 (Doing Clarence a Bit of Good), m.09gs5hy (Rallying Round Old George), m.09gs5hr (Absent Treatment), m.010r9gwy (Disentangling Old Duggie), m.010rlht0 (The Test Case), m.010rlh9d (Concealed Art) ]

                                                                 ✓ Real KB results — 7 entities

**Step 3**  Think: *I have a list of short stories that feature Reggie Pepper, including "Doing Clarence a Bit of Good." Now, I need to filter this list to only include stories that are of the type "short story" and not "Doing Clarence a Bit of Good" itself.* ✓ Plans type constraint

Action: Merge [ expression2 | book.short_story ]

Info: result_mid_list: [ m.09gs5hv (Helping Freddie), m.09gs5j0 (Doing Clarence a Bit of Good), m.09gs5hy (Rallying Round Old George), m.09gs5hr (Absent Treatment), m.010r9gwy (Disentangling Old Duggie), m.010rlht0 (The Test Case), m.010rlh9d (Concealed Art) ]

                                                                 ✓ Type verified

**Step 4 (Final)**  Think: *The merged results now confirm all short stories featuring Reggie Pepper. Since the question asks for stories that share a character with "Doing Clarence a Bit of Good," I should exclude the story itself from the final answer and report the remaining six.* ✓ Reasons about exclusion

Answer: m.09gs5hv m.09gs5hy m.09gs5hr m.010r9gwy m.010rlht0 m.010rlh9d

---

pred_answer: 6 MIDs (all correct)                                 ✓ **Correct (F1 = 1.0)**

---

**Key Insight.** This case study illustrates that **grounded environment feedback is the decisive factor**. Without it, KBQA-o1's MCTS search is "flying blind"—the policy model generates plausible-looking action sequences, but neither the policy nor the reward model ever observes actual KB execution results. This makes errors like wrong relation directions impossible to detect. KBQA-R1's multi-turn interaction paradigm enables genuine closed-loop reasoning that is both more accurate and dramatically more efficient.

