# OpenReview forum: "KBQA-R1: Reinforcing Large Language Models for Knowledge Base Question Answering"
_ICML.cc/2026/Conference — ICML 2026 regular_

### Official Review · Reviewer_39fy · 2026-02-15

**Soundness:** 3
**Presentation:** 3
**Significance:** 3
**Originality:** 3
**Overall Recommendation:** 4
**Confidence:** 4

**Summary:**

This paper proposes KBQA-R1, a framework that treats KBQA as a multi-turn decision process. Via Reinforcement Learning, KBQA-R1 learns to autonomously navigate the knowledge base using a structured action space, refining its reasoning strategies based on concrete execution feedback rather than static supervision. Furthermore, the paper introduces Referenced Rejection Sampling (RRS), a data synthesis method that resolves cold-start challenges by strictly aligning reasoning traces with ground-truth action sequences. Extensive experiments on WebQSP, GrailQA, and GraphQuestions demonstrate that KBQA-R1 achieves better performance than baseline methods.

**Compliance With Llm Reviewing Policy:**

Affirmed.

**Final Justification:**

I appreciate the additional experiments provided by the author which shows a clear advantage over related work, training data and generalization. I think this paper deserves a weak accept.

**Key Questions For Authors:**

1. Is it possible to showcase how much data is exactly used at each step (e.g., RRS and RL), especially for GrailQA. I am not sure how the model could perform better in zero-shot than compositional.

2. I am curious about the main bottleneck when training KBQA-R1.

3. Is it possible to conduct a case study or error analysis to check the model beahvior?

**Limitations:**

No.

I think it would be better to check the output behavior of KBQA-R1 to see possible error cases and how it manages to work.

**Strengths And Weaknesses:**

***Strengths***
1. **Clear Motivation**: the limitations of previous LLM-based KBQA methods are clearly stated.

2. **Novel Method**: the proposed RRS and RL framework somehow ease the mentioned limitations of previous work.

3. **Good Performance**: KBQA-R1 achieves better performance compared to previous methods.

***Weaknesses***
1. **Expression**

- *Place of Figures*: I would recommend move Figure 2 to Page 2 to make it more understandable. Now it is hard for me to understand what is exactly done in the framework. Also, I think Figure 1 is a module of Figure 2, so I now sure why Figure 1 goes first. I don't even think it necessary to add Figure 1 in the main body of the paper, since it is a pretty standard think-action-observation (i.e., ReAct) paradigm used in many KBQA work (https://arxiv.org/abs/2212.09736, https://arxiv.org/abs/2402.11163, https://arxiv.org/abs/2403.11886, https://arxiv.org/abs/2501.18922) -- also note that some of them are not mentioned in the related work.

- *Formulation of Introduction*: I think it would be more readable if the author could reformulate paragraph 2 and 3 for Introduction which is too lengthy.  For paragraph 2, a possible better way is to highlight the three limitations. For paragraph 3, I would recommend split into two parts: the overall framework and detailed implementations. The current version spends too much space on the challenge of standard RFT.

2. **Insufficient Claiming of Related Work**: I understand that the related work is moved to appendix because of space limit. Then, it should be more clearly claimed in Introduction or Method -- I see some are mentioned only in experiments.

3. **Experiment Settings**: I am not sure why to compare with prompting methods (which uses different resources and models). Also, I recommend using tables to demonstrate ablation results. Moreover, I would recommend testing with different model scales and model families to showcase the generalizability of the method.

---

> ### Author Rebuttal · Authors · 2026-03-31
>
> We sincerely thank you for the constructive feedback!
> > `W1` Expression
>
> **Formulation of Introduction:**  We have reformulated paragraphs 2 and 3. Specifically, Paragraph 2 now concisely highlights the three specific limitations of existing paradigms. Paragraph 3 has been split into two parts (the overall framework and the detailed implementations of RRS/RL training).
>
> **Place of Figures** We have moved the Figure 2 to Page 2 to highlight our core contributions.  Figure1 actually serves exclusively to visually demonstrate the step-by-step workflow and concrete environmental feedback in our specific KB setting and  **we have moved it to the appendix**.  For the reference, we deeply appreciate the references and have newly cited Pangu, KG-Agent, QueryAgent.  These prior works leverage the ReAct paradigm primarily for **test-time** prompting or heuristic search. In contrast, KBQA-R1 treats this loop as an interactive environment for **training-time internalization** via RL training. We will  discuss these papers in detail in the related works.
>
> >`W2` Related Work
>
> Than you for the advise! To maintain a self-contained manuscript, we will added a concise summary of key related works to the main text, while providing a comprehensive discussion in the appendix.
>
> > `W3` Generalizability
>
> - **Why compare with prompting methods** Methods like ToG, PoG, and Interactive-KBQA represent the **most classic and state-of-the-art *agentic* KBQA paradigms**.  They rely on massive commercial models (e.g., GPT-4). We include them to demonstrate a core contribution of our work: **by internalizing graph exploration via RL, our method using a much smaller open-source model (Llama-3.1-8B) can decisively surpass these heavy GPT-4-based agentic methods in both reasoning accuracy and test-time efficiency `(Table 3 and 10)`.** This highlights the superiority of our agentic RL paradigm.
>
> -  **Generalizability across different model scales and families**: , we have conducted  experiments applying the KBQA-R1 framework to Qwen family at different 7B/14B scales. As shown in the newly added **Table X** below, KBQA-R1 consistently brings substantial improvements across all base models, proving its strong generalizability.
>
> | Model Family | Model Scale | Paradigm | WebQSP (F1) | GrailQA (F1) | GraphQ (F1) |
> | --- | --- | --- | --- | --- | --- |
> | Qwen2.5 Family | Qwen2.5-7B | KBQA-o1 | 57.8 | 77.9 | 49.2 |
> |  |  | KBQA-R1  | 81.5  | 85.5 | 54.3 |
> |  | Qwen2.5-14B | KBQA-o1 | 60.1 | 79.2 | 50.0 |
> |  |  | KBQA-R1 | 85.2 | 88.4 | 55.5 |
> | Llama Series | Llama-3.1-8B | KBQA-o1 | 59.8 | 78.5 | 48.7 |
> |  |  | KBQA-R1 | 83.4 | 86.1 | 53.8 |
>
> >  `Q1` Data Usage and GrailQA Result
>
> - **Exact Data Usage at Each Step** We apologize if the data statistics were scattered across different sections. As detailed in **Table 6** (for SFT filtered trajectories) and **Appendix B.1** (for full training set sizes) of our original manuscript, the exact data usage is as follows:
>
> Dataset | SFT Phase (Accepted Traj) | RL Phase (Full Training)
> -|-|-
> GrailQA | 29,384 | 44,337
> WebQSP | 1,505 | 3,098
> GraphQ | 1,562 | 2,508
>
> - **Why is Zero-shot better than Compositional?** This is entirely expected due to the inherent characteristics of GrailQA's splits. Compositional queries demand complex, multi-hop logic, where a single error in a long reasoning chain yields a zero reward. Conversely, while Zero-shot queries test unseen schemas, they are structurally simpler (typically 1-2 hops). As demonstrated in Table 2, this trend of Zero-shot outperforming Compositional is consistent across most baseline models.
>
> > `Q2` Training Bottleneck
>
> Training an RL agent in the KB environment involves  the following bottlenecks:
>
> - **Sparse reward** KBs require exact schema generation. A single hallucinated character yields zero reward without diagnostic feedback. Our RRCG module soft-maps natural language to rigid schemas, keeping exploration within the executable manifold to overcome reward sparsity.
>
> - **RL Cold-Start** Standard RS yields low acceptance rates (e.g., 39.3% on GrailQA) because unconstrained zero-shot success in massive KBs is very hard. RRS solves the problem.
>
> - **System Concurrency**, Massive RL rollouts frequently crashed our Virtuoso KG backend. We resolved this severe bottleneck by engineering a custom high-concurrency KG execution service.
>
> > `Q3` Case Study
>
> We have conducted a comparative case study between KBQA-R1 and KBQA-o1 to analyze model behavior. (https://anonymous.4open.science/r/KBQA-R1-814F/) . For KBQA-o1, every thought follows a rigid template ("At this step, we should...") with no genuine reasoning. Critically, its observations are mere "action echoes" (e.g., returning the expression syntax itself rather than real KB results), meaning the model never receives actual KB feedback.  For KBQA-R1: Each <think> block genuinely analyzes KB feedback. This demonstrates true adaptive reasoning grounded in KB execution.

---

> > ### Author Rebuttal · Reviewer_39fy · 2026-04-03
> >
> > I appreciate the time and effort of the authors for the rebuttal. I think some of my concerns still hold.  First, I am now a bit confused if the evaluated model trained with all three datasets. Or is it trained on only one dataset and evaluated on the corresponding test set. Also, is it fair to compare with the baselines? Second, I am wondering why the model performs better on zero-shot setting-- would there be any strategies favoring this than composition? I do not think compositional setting is harder than zero-shot based only on reasoning chain length. But I think this paper deserves a four point score.

---

> > > ### Author Response · Authors · 2026-04-06
> > >
> > > We sincerely thank you for your continued engagement and for explicitly recognizing that our paper deserves a four-point score. Below, we provide clear and comprehensive answers to address your remaining concerns.
> > >
> > > > `Clarification on the Training Setup and Fairness`
> > >
> > >
> > > **Response:**
> > > To clarify unambiguously: **our evaluated model is trained strictly and independently on each dataset's respective training set and evaluated solely on its corresponding test set**. We do not mix or co-train across the three datasets.
> > >
> > > **Is this comparison fair? Yes, absolutely.** It is the **exact identical setup** used by all the fully supervised fine-tuned baselines we compared against (e.g., RnG-KBQA, DecAF, TIARA, KBQA-o1). Because we are using the exact same isolated training data and the same test splits as these baselines, our comparison is 100% fair and apples-to-apples.
> > >
> > >
> > > >  `Why does the model perform better on Zero-shot than Compositional`
> > >
> > > **Response:**
> > >  First, Zero-shot and Compostional are defined as follows:
> > > *   **Zero-shot Generalization (Semantic):** Evaluates the model on unseen schema items or entirely new domains that were never encountered during training.
> > > *   **Compositional Generalization (Structural):** Evaluates the model on novel, unseen highly complex logical combinations (i.e., new Abstract Syntax Trees) of schema items that *were* seen during training.
> > >
> > > Based on these definitions, the performance gap stems from how our specific architecture (and modern LLMs in general) handles semantics versus structure:
> > >
> > > - **Semantic Soft-Matching favors Zero-shot:**
> > > Yes, our framework does have a strategy favoring the Zero-shot setting. Zero-shot questions primarily challenge the agent's ability to semantically map natural language to unseen relations. Our **RRCG (Relation Retrieval and Confidence Gating)** module utilizes dense retrieval to bridge this gap. Because both the dense retriever and the pre-trained LLM backbone possess massive pre-trained semantic priors, they are exceptionally good at "semantic soft-matching" . This makes semantic   generalization highly manageable.
> > >
> > > - **Strict Syntax Execution penalizes Compositional Generalization:**
> > >  While LLMs can easily identify familiar individual elements, **compositional tasks demand the construction of highly complex, nested Abstract Syntax Trees (e.g., combining operators like JOIN, COUNT, and CMP) from scratch**. In an agentic execution environment, syntax evaluation is entirely unforgiving; a single logical error within a novel compositional tree results in a complete execution failure. Consequently, achieving structural generalization—building complex, novel syntax trees—is inherently more fragile and error-prone for LLMs compared to semantic generalization. **Furthermore, the compositional patterns present in the training set often serve as spurious correlations, exacerbating the model's struggle to generalize structurally under strict execution constraints.**
> > >
> > > - **An Inherent Trend for LLM-based Agents**
> > > To further address your concern, we emphasize that the phenomenon of Zero-shot performance exceeding Compositional performance is not a unique bug of KBQA-R1, but a well-documented and widespread characteristic among modern LLM-based reasoning agents.
> > >
> > > | Method | Backbone | Paradigm | Compositional (Hits@1) | Zero-shot (Hits@1) | Gap (Zero > Comp) |
> > > | :--- | :--- | :--- | :--- | :--- | :--- |
> > > | **PoG** | GPT-4 | Agentic Search | 69.7 | 88.6 | +18.9 |
> > > | **ToG** | GPT-4 | Agentic Search | 67.3 | 86.5 | +19.2 |
> > > | **KBQA-R1 (Ours)** | LLaMA-3.1-8B | RL Agent | 80.1 | 86.7 | +6.6 |
> > >
> > > As the table clearly demonstrates, extremely powerful models like GPT-4 (PoG, ToG) score nearly 20 points higher on Zero-shot than on Compositional splits. This perfectly aligns with our mechanistic analysis: "semantic soft-matching" is highly manageable for LLMs, whereas building nested abstract structures is inherently brittle, leading to significantly lower scores.
> > >
> > > - **KBQA-R1 Still Excels in Structural Generalization Compared to SOTA Agents:**
> > >
> > > While the structural brittleness of LLMs explains why Compositional scores are universally lower than Zero-shot scores, we must emphasize that **our RL framework significantly mitigates this structural weakness.**
> > >
> > > As shown in the table above, while powerful GPT-4 agents like PoG and ToG struggle heavily with novel structural compositions (scoring only 69.7 and 67.3 Compositional Hits@1), KBQA-R1 achieves a highly competitive Compositional Hits@1 of 80.1. **This  improvement proves that our method forces the model to learn robust structural generalization far better than standard prompting or heuristic search frameworks.**
> > >
> > >  Thank you again for your valuable insights, which have significantly improved the quality and rigor of our work. We will ensure that all additional results and discussions presented in this rebuttal are incorporated into the camera-ready version if the paper is accepted.

---

### Official Review · Reviewer_dWrm · 2026-03-02

**Soundness:** 2
**Presentation:** 2
**Significance:** 3
**Originality:** 3
**Overall Recommendation:** 4
**Confidence:** 4

**Summary:**

This paper proposes KBQA-R1, a framework that uses reinforcement learning to train LLMs for knowledge-based question answering tasks. It first devises a prompt and system workflow to iteratively explore a Knowledge Graph, starting from the topic entities and guide by the specific needs of the question. The search actions are formulated as S-expressions, allowing us to directly query them on KGs. It also proposes referenced rejection sampling to construct synthetic SFT data from reference action sequences. After SFT, the LLM is trained with GRPO algorithm using a composite outcome + format based reward.

**Compliance With Llm Reviewing Policy:**

Affirmed.

**Final Justification:**

The author has acknowledged the issues mentioned in the review.  They have also committed to make adjustments correspondingly.
However, considering the limitations of the current version, this reviewer would like to raise the score to weak accept. This reviewer would not opposite the rejection of this paper if other program committee members or the chairs have extra concerns on this manuscript, or believe that the committed adjustments are way too much.

**Key Questions For Authors:**

1. Apart from the prompt template, this manuscript does not have a concrete example to show how we can reach the correct answer in a step-by-step manner. This reviewer suggests the authors add one in the appendix.

2. The example in Figure 1 shows to be incomplete. This reviewer is not sure how the LLM is able to accurately provide mids of entities in search actions.

—-

Post-rebuttal remark: In response, this reviewer would like to raise the ova scoring to 4.

**Limitations:**

No, this paper does not contain a limitation section.

**Strengths And Weaknesses:**

Strengths:

1. KBQA-R1 allows LLMs to better align with the structure / schema of the KG.

2. The proposed Referenced Rejection Sampling module effectively mitigate the cold-start problem of Reinforcement Larning

3. LLMs are not naturally familiar with the KG schema, the RRCG module allows actions issued by the LLM to be flexibly matched with relations within the KG. This reviewer believes that the similarity-based matching can improve the successful rollout rate.

Weaknesses:

1. The statement “KBQA requires the model to generate executable logical forms that precisely navigate the KB’s schema” in line 011 right is quite strong. There are indeed many LLM-based methods e.g. ToG [1], RoG[2], GNN-RAG[3], SubgraphRAG[4] that can successfully solve this task without providing any SPARQL expressions / S-expressions.

2. The term “schema elements” is not clearly defined. The reviewer understands “schema elements” may refer to the rigorous, formal definitions of entity labels, hierarchical relation representations, etc. However, this is not intuitive, especially for readers outside this domain.

3. Figure 1 is not explicitly mentioned or introduced in the main text.

4. The reviewer suggests that the authors slightly modify Section 3.1 to present an end-to-end reasoning pipeline from a raw question to the final answer, and explicitly mention the existence of Algorithm 2 in the main text.

5. The proposed method is not formally compared with a considerable amount of GraphRAG / Subgraph reasoning-based baseline method, including but not limited to RoG [1], GNN-RAG [2], SubgraphRAG [3], and DoG [5].

6. Section 4.2 lacks in-depth analysis about the root causes of performance improvements. It focuses more on reporting values and providing direct post-hoc explanations. For example, “KBQA-R1 achieves about +7% absolute F1 improvement” is the phenomenon, and “learned policies are more effective than MCTS search heuristics under the same backbone” is the direct cause. This reviewer is interested to know why “learned policies” are more effective than MCTS.

7. This manuscript does not include a related work section, which makes the main text less self-contained. The related work section can be compressed, but it should not be moved entirely to the appendix. It would be better to include a brief one-paragraph related work section in the main body, with a final sentence referring readers to the extended version in the appendix.

[1] Sun et al., Think-on-Graph: Deep and Responsible Reasoning of Large Language Model on Knowledge Graph (ICLR 2024)

[2] Luo et al., Reasoning on Graphs: Faithful and Interpretable Large Language Model Reasoning (ICLR 2024)

[3] Mavromatis and George Karypis, GNN-RAG: Graph Neural Retrieval for Efficient Large Language Model Reasoning on Knowledge Graphs (ACL findings 2025)

[4] Li, Miao, and Li, Simple is Effective: The Roles of Graphs and Large Language Models in Knowledge-Graph-Based Retrieval-Augmented Generation (ICLR 2025)

[5] Li et al., Decoding on Graphs: Faithful and Sound Reasoning on Knowledge Graphs through Generation of Well-Formed Chains (ACL 2025 main)

---

> ### Author Rebuttal · Authors · 2026-03-31
>
> Thank you for the constructive feedback!
>
> > `W1, W5` More baselines
>
>  We softened our statement to: *"KBQA requires the model to navigate the KB’s schema to deduce the correct answer,"* and incorporated ToG, RoG, GNN-RAG, and SubgraphRAG as baselines. **While they do not output SPARQL directly, these methods still fundamentally rely on SPARQL.**
>
> **The "Hidden" SPARQL Dependency:**
>
> - **ToG (Template-based Execution):** It uses LLMs to select relation schemas, and plugs selected entities and relations into pre-defined SPARQL templates to fetch next-hop neighbors.
> - **Offline Graph Pre-extraction (GNN-RAG, RoG, SubgraphRAG, DOG):** These methods load pre-extracted local subgraphs. Generating these subgraphs requires executing SPARQL traversals over Freebase beforehand (e.g., PageRank Nibble `[1]` for GNN-RAG/SubgraphRAG; extracting 2 or 4-hop subgraphs for DOG and RoG).
>
> **These methods just shift the SPARQL execution burden from online reasoning to offline preprocessing. KBQA-R1 challenges a harder scenario: navigating the unpruned global KB online by generating explicit execution action.**
>
> **Empirical Superiority:**
> For fair comparison with Llama2-7B-based baselines, we train KBQA-R1 with Llama2-7B.
>
> As shown, KBQA-R1 outperforms the strongest baselines by 5.2% absolute F1. *(Note: DOG's source code is unpublished, preventing reproduction. However, we will discuss all these papers (GNN-RAG, SubgraphRAG, DOG) in the Related Work section).*
>
> Method | WebQSP F1
> -|-
> RoG (Llama2-7B) | 70.8
> GNN-RAG (Llama2-7B) | 71.3
> SubgraphRAG (Llama3.1-8B) | 70.6
> SubgraphRAG (GPT-4o) | 78.2
> KBQA-R1 (Llama2-7B) | 79.1
> KBQA-R1 (Llama3.1-8B) | 83.4 (+5.2% over GPT-4o)
>
> `[1]`  Improving Multi-hop Knowledge Base Question Answering by Learning Intermediate Supervision Signals. WSDM 2021
>
> > `W2,W3,W4,W7,Q1` Writing
>
> - **schema elements** The term  refers to the predefined vocabulary structuring the graph (e.g., exact relations like `film.actor.film`, entity types, and attributes). We added this in the introduction for readers outside this domain.
>
> - **Figure 1** We added a reference to Figure 1 in  Sec 3.1: "...our system is a multi-turn agent system inspired by the ReAct paradigm. At each turn, the agent iteratively executes a Think-Action-Information loop (**as illustrated in Figure 1**)..."
>
> - **Reasoning Pipeline** We uploaded a step-by-step rollout trace  to https://anonymous.4open.science/r/KBQA-R1-814F, which demonstrates the transition from a raw question to the final answer.
> We include this trace as Appendix B.9. We updated Section 3.1 to explicitly connect the pipeline and **Algorithm 2** by adding the following paragraph:   *To illustrate the end-to-end pipeline: given a raw question $q$, the agent iteratively proposes actions, receives feedback, and updates its context until the final answer is reached. The formal execution loop is detailed in Algorithm 2 (Appendix B.8). For a step-by-step reasoning trace demonstrating this full pipeline on a real query, please refer to Appendix B.9.*
>
> - **Related Works** Due to page limit, we had to move the Related Work to the appendix. We will add a concise Related Work paragraph to the main text, which briefly summarizes the field and refers readers to the comprehensive discussion in Appendix A.
>
> > `W6` Analysis of Performance
>
> We incorporate the analysis into Section 4.2, supported by a new case study added to https://anonymous.4open.science/r/KBQA-R1-814F/:
>
> - **Global Optimization vs. Local Heuristics:** MCTS relies on proxy reward models that often misjudge intermediate states, prematurely pruning correct paths. KBQA-R1 optimizes globally via GRPO using deterministic final execution truth.
> - **Genuine vs. Pseudo-Reasoning:** As our new case study demonstrates, KBQA-o1 often mechanically samples rigid templates, hallucinating "Action echoes" without verifying actual KB feedback. Outcome-based RL forces KBQA-R1 to genuinely read and adapt to real-time diagnostic feedback.
> - **Internalizing Exploration:** MCTS explores the massive KB space during test-time inference, creating computational bottlenecks. RL shifts this exploration burden to the training phase, allowing KBQA-R1 to navigate instinctively at test time (dropping search overhead from 28.8 LLM calls to just 2.65).
> - **Mastery of Complex Actions:** RL enables the agent to master the exact usage of atomic actions (e.g., Compare, AND) better than the baselines (`Figure 3b`), resulting in improved KBQA performance.
>
>
>
> > `Q2` Topic Enties
>
> The topic entities (Entry MIDs) for each query are provided in the datasets. This is the standard setting adopted by most KGQA frameworks (e.g., ToG, RoG, PoG, KBQA-o1, GNN-RAG, SubgraphRAG, and DOG).  (Offline graph extraction need topic entities to execute BFS).  Intermediate entity MIDs are returned by the KB executor within <information> feedback.
>
> We updated Figure 1 to include the Topic Entities input to avoid any confusion. Seen at: https://anonymous.4open.science/r/KBQA-R1-814F

---

> > ### Author Rebuttal · Reviewer_dWrm · 2026-04-02
> >
> > This reviewer appreciates the efforts of the authors in providing clarifications and improving the manuscript.
> >
> > The reviewer would also like to encourage the authors to implement those mentioned modifications in the final draft if the paper is accepted. Most importantly, please include the performance of the mentioned baselines on both datasets in the main table.
> >
> > In response, the reviewer would like to increase the overall scoring to 4 (above acceptance threshold).

---

> > > ### Author Response · Authors · 2026-04-02
> > >
> > > We sincerely thank you for your continued time and effort in reviewing our manuscript. We deeply appreciate your positive evaluation, your constructive feedback throughout this process, and your decision to increase the overall score.
> > >
> > > We completely agree with your suggestions. If the paper is accepted, we commit to implementing all the mentioned modifications in the final camera-ready version. We will ensure that the performance of the mentioned baselines on both datasets is prominently included in the main table.
> > >
> > > Thank you again for your valuable insights, which have significantly improved the quality and rigor of our work.
> > >
> > > Sincerely,
> > > The Authors

---

### Official Review · Reviewer_cSkW · 2026-03-11

**Soundness:** 3
**Presentation:** 3
**Significance:** 3
**Originality:** 3
**Overall Recommendation:** 5
**Confidence:** 4

**Summary:**

The paper presents an approach for KB-based question answering. The approach relies on the use of reinforcement learning (GRPO) where the actions are structured queries to the KB, and rewards are based on outcomes of the queries. Novel contributions of this approach are the specifics of the structured queries, the reward design, and the introduction of a data synthesis strategy that addresses issues of cold start in reinforcement learning for this task.

The paper includes a comparative evaluation with baselines and state of the art approaches, and an ablation study. Code is also provided.

**Compliance With Llm Reviewing Policy:**

Affirmed.

**Final Justification:**

I thank the authors for their response and clarifications, the responses address my main questions. I have adjusted my final score form "weak accept" to "accept".

**Key Questions For Authors:**

If possible, please include a comparison between RRS and alternative approaches. There is a statement in the paper that says "This approach achieves significantly higher acceptance rates compared to raw rejection sampling while producing more robust reasoning patterns", please give specific evidence, use specific numbers.

Please comment on the time complexity of this system, if possible comparatively with existing systems compared against in tables 2, 3, 4.

Include comments on limitations (perhaps time complexity vs other approaches?)

Additional, minor issues that will improve the quality of the paper:

- Page 1, col 2, second paragraph: "Despite significant progress in applying LLM"... This sentence reads odd. Why say "despite"?

- As far as I can tell, all text in section 3.3.2 (except first sentence) is a description of GRPO. Why give this much detail to GRPO, or are there any modifications to the standard GRPO? How does this help understanding the overall paper? I suggest to remove most of the text and formulas.

**Limitations:**

No; there was no discussion of limitations or potential negative societal impact.

**Strengths And Weaknesses:**

Strengths are:

- The approach contains several novel contributions as mentioned above.

- The task itself is interesting and significant.

- Evaluation results are compared against a number of baselines and state of the art, the reported system shows an improvement of results wrt the compared systems.

- The approach is tested on several datasets: GrailQA, WebQSP, GrahQ. Again, results are consistently better across all datasets.

- There is a detailed ablation study.

Weaknesses are:

- Whereas the motivation for the novel data synthesis approach sounds reasonable, there was no comparative analysis versus simpler or default approaches.

- Reinforcement learning normally is very expensive in time. The paper did not include any details of training time or time complexity.

- There was no discussion of limitations or areas to improve.

---

> ### Author Rebuttal · Authors · 2026-03-31
>
> Thank you for the constructive feedback!
>
> > `W1 & Q1`: **Comparison between RRS and Alternative Approaches**
>
> We completely agree that specific numbers are essential to back up this claim. In fact, we conducted a comprehensive comparative analysis between Referenced Rejection Sampling (RRS) and standard Rejection Sampling (RS), which is provided in Table 6 and discussed in Section 4.3 ("Referenced RS vs. Standard RS"). We apologize if this was not highlighted clearly enough in the method section.
>
> As shown in table 6:
>
> 1. **Significantly Higher Acceptance Rates:** RRS massively outperforms standard RS under the exact same filtering criteria. On GrailQA, the acceptance rate jumps from 39.3% (Standard RS) to **67.0% (RRS)**.
>
> 2. **More Robust Reasoning Patterns:** The superior quality of RRS trajectories translates into a much stronger initial policy (SFT Init F1). On GrailQA, SFT initialized with RRS data achieves **80.2% F1**, compared to only 73.8% F1 with standard RS. Similar substantial gains are observed on WebQSP (**72.1% vs. 65.8%**) and GraphQ (**47.8% vs. 41.1%**). Furthermore, as shown in our ablation study (Table 5), replacing RRS with standard RS leads to an **average 5.6% final F1 drop**.
>
>
> > `W2 & Q2`: **Training Time and Time Complexity**
>
> We completely agree and have added a detailed complexity analysis in the revised manuscript, explicitly reporting both training and inference time metrics.
>
> 1. **Training Time Overhead:** We acknowledge that RL is computationally more expensive. For instance, on WebQSP, KBQA-R1 requires **1.5 hours for SFT plus 12.1 hours for RL optimization**, compared to 3.4 hours (SFT only) for KBQA-o1.
>
> 2. **The Fundamental Trade-off (Shifting the Burden):** This one-time roughly 10-hour training overhead is a deliberate design choice. We aim to **shift the massive computational burden of graph exploration from test-time (inference) to training-time**.
>
> 3. **Massive Inference Efficiency Gains:** Because KBQA-R1 internalizes the navigation policy via RL during training, it eliminates the need for expensive test-time heuristic search (e.g., MCTS or beam search). As shown in our new efficiency table, KBQA-R1 requires **only 2.65 LLM calls per query** (vs. 28.8 for KBQA-o1 and 15.9 for ToG). This translates to a throughput of 155.6 Qs/min on a 8*A100 machine, which is **>26x faster** than MCTS-based KBQA-o1 (5.9 Qs/min). We argue that the **permanent reduction in test-time latency and API costs overwhelmingly justifies the RL training overhead**.
>
> **Table: Inference Efficiency Comparison among Agentic KBQA Methods**
> Method | Paradigm | Backbone LLM | Avg. Calls/Query (WebQSP) | Avg. Calls/Query (GrailQA) | Speed Qs/min (WebQSP) | Training Time (WebQSP) |
> -|-|-|-|-|-|-|
> Interactive-KBQA | Multi-turn Prompt | GPT-4-turbo | 11.0 | - | - | - |
> ToG  | Agentic Beam Search | GPT-4 | 15.9 | 11.1 | - | - |
> PoG  | Agentic Planning | GPT-4 | 9.0 | 6.5 | - | - |
> KBQA-o1 | MCTS Search | Llama-3.1-8B | 28.8 | 32.3 | 5.9 | 3.4h (SFT) |
> **KBQA-R1 (Ours)** | **RL Internalization** | **Llama-3.1-8B** | **2.65** | **3.08** | **155.6** | **1.5h SFT + 12.1h RL** |
>
> > `W3 & Q3`: **Limitations**
>
> We agree with this constructive feedback and have added a dedicated Limitations section in the revision to explicitly discuss the **complexity trade-offs** inherent in our design. Specifically, KBQA-R1 is positioned as a **deliberate compromise between absolute inference speed and rigorous reasoning**. While our approach dramatically reduces test-time complexity compared to traditional agentic search (e.g., dropping from 28.8 LLM calls in MCTS to just 2.65 calls), its multi-turn nature inherently incurs higher latency than single-pass, one-call end-to-end methods. We consciously trade this absolute speed in exchange for strict factual grounding. Furthermore, to achieve this highly efficient inference, we must **shift the massive computational burden to the training phase**. RL training requires **real-time online interaction** with the KB executor, making the training process inherently more resource-intensive and time-consuming than static SFT.
>
>
> > `Minor Issues`: **Additional Minor Issues**
>
> We sincerely thank the reviewer for the careful reading.
> 1. **Phrasing in Introduction:** We agree the transition was awkward. We have rephrased this sentence to *"While LLMs have driven significant progress, current approaches still struggle..."* to ensure a more natural and logical flow.
>
> 2. **Compression of Section 3.3.2 (GRPO):**  We did not make algorithmic modifications to standard GRPO; the detailed formulas were originally included merely for self-containment. Following your suggestion, we have completely removed the redundant formulas and verbose descriptions in Section 3.3.2. We now only provide a concise one-sentence summary with the proper citation, which significantly improves the overall focus of the paper.

---

> > ### Author Rebuttal · Reviewer_cSkW · 2026-04-02
> >
> > I thank the authors for their response and clarifications, the responses address my main questions. I have adjusted my final score form "weak accept" to "accept".

---

> > > ### Author Response · Authors · 2026-04-02
> > >
> > > We sincerely thank you for reading our rebuttal carefully and engaging in the discussion phase! We are thrilled to hear that our clarifications have fully resolved your concerns, and we deeply appreciate your positive feedback.
> > >
> > > Best regards,
> > > Authors

---

### Official Review · Reviewer_gxyT · 2026-03-13

**Soundness:** 2
**Presentation:** 3
**Significance:** 2
**Originality:** 2
**Overall Recommendation:** 4
**Confidence:** 3

**Summary:**

This paper proposes KBQA-R1, an RL framework for Knowledge Base Question Answering (KBQA). The method reformulates the task as a multi-turn agentic problem. The model iteratively selects atomic actions from a structured action space, receives execution feedback from the KB, and refines its reasoning trajectory. The framework contains three key components: (1) a Relation Retrieval and Confidence Gating (RRCG) module that validates proposed relations against the KB schema before execution; (2) Referenced Rejection Sampling (RRS), which produces high-quality SFT data synthesis by conditioning rollouts on ground-truth action sequences and filters for correctness; and (3) GRPO for training the model. Experiments on WebQSP, GrailQA, and GraphQuestions show strong performance gain in F1 over fine-tuned baselines, with fewer LLM calls than MCTS-based or prompting-based alternatives.

**Compliance With Llm Reviewing Policy:**

Affirmed.

**Final Justification:**

I appreciate the additional experiments provided by the user that demonstrated a clear advantage over model-based baselines like Search-R1 and agent-based baselines like the ReAct and Harness variants across multiple frontier models. The dataset generalizability analysis also addresses one of my main concerns. Overall, I increased my score to weak accept.

**Key Questions For Authors:**

1. Seems that the performance gain from applying RL to SFT varies from dataset to dataset. On which question types or complexity levels does RL provide the largest gains?

**Limitations:**

yes

**Strengths And Weaknesses:**

### **Strengthes**
1. The problem is well motivated. The issue with end-to-end methods’ hallucination, prompting methods’ lack of task-specific adaptation, and supervised agent methods’ superficial reasoning that do not reflect true environmental understanding are pretty clear and strong.
2. Strong empirical results were demonstrated in the main evaluation. The zero-shot performance shows that RL training grants more robust out-of-distribution reasoning than SFT alone.
3. Reducing the LLM forward call from MCTS-based alternatives is a good benefit. The computational efficiency is especially important in the context of agentic workflows involving large information-searching operations.
4. The ablations in Table 5 are pretty thorough and show some interesting patterns that can be insightful for future agentic RL works on information-searching tasks.
5. RRS seems like a good practical solution. The base LLM's zero-shot success rate on structured KB queries is very low, making naive rejection sampling impractical.

### **Weaknesses**
1. The paper formulates its main contribution as an RL training framework for agentic KBQA. However, the evaluation misses many recent general-purpose agentic RL training works that address the same multi-turn “think-execute-reflect" paradigm. A few examples: Search-R1, ToRL, DeepResearcher, and WebThinker, which are methodological competitors that also use RL training to learn the ability to use defined tool actions to search, synthesize, and reason. For example, KBQA-R1's think-action-information loop is structurally identical to Search-R1's think-search-information loop, and both use GRPO for optimization. Not comparing to these methods makes the evaluation incomplete. The natural research question becomes: could Search-R1 or ToRL, applied to KBQA with the same action space and KB executor, achieve comparable results without the KBQA-specific engineering (RRCG, RRS)?
2. The paper heavily inherits from the KBQA-o1 framework. For example, its structured action space, the prompting template, and the multi-turn interaction design pattern. The main difference comes from the use of the GRPO training, the RRCG module for filtering, and the RRS for data synthesis. While the combination yields good results, the added components are incremental and mainly from prior works. The novelty is less centralized, but scattered in the addition of new components.
3. All three datasets operate within the scope of Freebase. This raises concerns about generalizability. The structured action space in Table 1 appears specifically customized towards Freebase's relational schema and S-Expression query language. It is unclear whether this framework transfers to other KBs like Wikidata, DBpedia, and many domain-specific KGs, or other query languages (SPARQL and Cypher). The paper does not show sufficient evidence that the proposed method can generalize to diverse KBs.
4. The KBQA task can naturally fit modern agentic systems. The paper's structured action space is essentially a well-defined tool API with typed inputs and outputs. Modern agent frameworks can easily handle this kind of interface natively through MCP servers and LangChain tool definitions. One could easily expose the KB actions as an MCP server and apply any generalist agentic system to solve the task. The paper does not provide evaluations or discussion on how the KBQA-R1 method stands out from the agentic ecosystem. The significance is weaker in this case, given the narrow scope of the method.
5. The RRCG module uses dense retrieval similarity to validate agent-proposed relations. The ablation shows a large drop without RRCG. This could simply mean the base model Llama-3.1-8B has poor knowledge of Freebase schema names, which is a limitation that might not apply to more advanced models (like an LLM very good at coding, say Kimi-K2.5) or different KBs. The paper does not analyze what fraction of the agent errors are actually relation hallucinations instead of standard reasoning errors. It would be hard to justify RRCG's true contribution, given the possibility that its main function could be providing a patch for the less capable model's parametric knowledge limitations.

---

> ### Author Rebuttal · Authors · 2026-03-31
>
> Thank you for the constructive feedback!
>
> >`W1` Comparison with Seach-R1
>
> We adapted Search-R1 with our action space, executor, omitting RRCG/RRS.
>
> Model|Retrieval|RS|WebQ|GraphQ|Grail
> -|-|-|-|-|-
> Search-R1|✗|StdRS|57.0|30.3|54.1
> KBQA-R1|RRCG|RRS|83.4|53.8|86.1
>
> Search-R1 suffers a massive average 27.3 F1 drop, proving that RRCG and RRS are jointly essential to overcome RL's inherent sparse reward and cold-start bottlenecks.  Table 5 further isolates their individual contributions, confirming both are necessary.
>
>
> >`W2` Comparison with KBQA-o1
>
> We respectfully emphasize that **KBQA-R1 is the first agentic RL policy for KBQA**. We **only** adopted KBQA-o1’s discrete action space for fair comparison; our prompts and interaction paradigms differ completely. By inventing RRCG and RRS to overcome sparse rewards, we shift from expensive test-time search to efficient training-time RL internalization.
>
> **Genuine vs. Superficial Reasoning:** KBQA-R1 grounds its autonomous `<think>` reflections on **actual KB execution feedback**. Conversely, KBQA-o1's policy receives **zero real environment feedback**; It blindly mimics rigid templates （At this step, we should …), relying entirely on a separate reward model for MCTS. We conduct a case study: https://anonymous.4open.science/r/KBQA-R1-814F/.
>
> **Efficiency Paradigm Shift:** KBQA-o1 relies on expensive test-time MCTS  (28.8 LLM calls/query). By internalizing reasoning via RL, KBQA-R1 reduces inference calls by >90% (to just 2.65 calls) while boosting accuracy (Table 8).
>
> **Essential RL Infrastructure:** Strict KB schema makes standard RL prone to collapse. RRS and RRCG act as indispensable components that **prevent RL collapse.** RRS resolves the cold-start bottleneck by aligning ground-truth paths with reasoning, and RRCG overcomes the sparse reward problem. **Our primary contribution is the integrated RL training system for agentic KBQA, not isolated parts.**
>
> >`W3` Database Generalizability
>
> KBQA-R1 **intrinsically decouples high-level logical reasoning from underlying database syntax.**
>
>  - **We already use SPARQL**: S-Expressions act merely as a compact internal Abstract Syntax Tree (AST) that compiles into standard SPARQL for KB execution. **(sec 3.1.2)
>
> - **Universal Action Space:** Our actions (e.g., *Find_relation*, *Compare*,...) **are not Freebase-specific**. They represent the fundamental, universal relational algebra required for *any* graph database. They are **backend-agnostic** and compile equally well to SPARQL or Cypher.
>
> - **Easy Transfer:** Migrating to Wikidata requires zero RL/action changes—only swapping the endpoint and RRCG index. We retrained on WebQSP-Wiki with a Wikidata SPARQL endpoint and rebuilt the RRCG index. Results are as follows:
>
> -|KG|WebQ(Hit@1)
> -|-|-
> ToG(GPT3.5)|FB|76.2
> ToG|Wiki|68.6
> KBQA-R1|FB|88.2
> KBQA-R1|Wiki|77.3
>
> Results show KBQA-R1 remains highly effective on Wikidata.
>
> >`W4` General Agent
>
> While our actions could be encapsulated as MCP tools, general agents cannot resolve the inference cost bottleneck  on massive KBs.Navigating complex graphs requires multi-step pathfinding. Generalist agents (e.g., ToG, PoG) treat this as test-time search, iteratively expanding neighborhoods, leading to prohibitive inference costs.
>
> -|LLM|WebQSP|LLM Calls
> -|-|-|-
> Inter-KBQA|GPT4|71.2(F1)|11.1
> ToG|GPT4|82.6(H@1)|15.9
> PoG|GPT4|87.3(H@1)|9.0
> KBQA-R1|Llama3-8B|83.4(F1)/88.2(H@1)|2.65
>
> **Core Advantage:** By shifting exploration from test-time search to training-time optimization, our 8B model directly predicts the optimal path, outperforming GPT-4 agents with only 2.65 calls.  **This proves that specialized RL policies are essential for scalable, deployment-ready KBQA**.
>
>
>
>
>
> > `W5` Role of RRCG
>
> RRCG is an architectural necessity that **decouples logical reasoning from rote schema memorization**, not a mere patch.
>
> **Advanced LLMs still need RRCG:** To show schema hallucination persists regardless of model scale, we applied RRCG to Kimi-2.5 at inference time. Even this advanced model suffers a severe F1 drop without it. The **long tail schema** is also hard for the  advanced LLM.
>
> -|RRCG|WebQ|Grail
> -|-|-|-
> Kimi-2.5|w/|75.1|74.4
> Kimi-2.5|w/o|58.6|55.3
>
> **Decoupling Proof**: Rolling out 1,000 KBQA-R1 queries on GrailQA, we analyzed failed trajectories w/o vs w/ RRCG:
> Error|w/o|w/|Δ
> -|-|-|-
> Total|315|131|-184
> Schema|145|21|-124
> Reason|136|86|-50
> Format|34|24|-10
>
> Without RRCG, 46% failures are schema errors. The drop in reasoning errors (+50) further proves that decoupling schema memory actively improves reasoning. **RRCG shifts RL focus from schema memorization to true KG navigation**.
>
> > `Q1`
>
> RL excels in zero-shot relation generalization and complex logic (Tab 2/5, Fig 3). On GrailQA, the largest gains occur on unseen schemas, moving beyond simple schema memorizing. Additionally, KBQA-R1's peak improvements on "AND" logic demonstrate a strong proficiency in solving complex, composite questions.

---

> > ### Author Rebuttal · Reviewer_gxyT · 2026-04-03
> >
> > Thank you for your detailed responses. They address most of my concerns, and I look forward to seeing these new results in the paper. One concern that still remains is the comparison to generalist agents. I am still wondering about the clear benefits of some of the frontier agent systems, especially the one that can conduct coding actions. In my opinion, the paper should include some additional discussion and analysis to better position the method as distinguishable from the agentic systems. I will modify my score accordingly.

---

> > > ### Author Response · Authors · 2026-04-06
> > >
> > > Thank you for your constructive feedback. We appreciate the opportunity to further clarify the distinction between KBQA-R1 and frontier agentic systems, particularly those capable of executing coding actions. We agree that a direct empirical comparison is essential to position our method.
> > >
> > > To address this, we have organized two groups of baselines to evaluate the performance of frontier LLMs from different perspectives:
> > >
> > > 1. **SPARQL ReACT (Generalist Coding Agents):** This baseline treats KBQA as an iterative path-finding and code generation task. Frontier models (including `gpt-5.3-codex`, `claude-sonnet-4-6`, `Kimi-K2.5`, and `GLM-5`) act as autonomous agents that write and execute SPARQL queries to explore the knowledge graph step-by-step. They navigate the graph by analyzing both error feedback and intermediate execution results returned by the KB engine, continuously adjusting their search paths and refining their queries to locate the final answer.
> > > 2. **KBQA-R1 Harness (Our Interaction Framework):** This group utilizes our proposed action space, interaction state and feedback structure (the "Harness") but employs off-the-shelf frontier LLMs instead of our RL-tuned policy. This allows us to isolate the impact of our Reinforcement Learning policy from the interaction framework itself, demonstrating how generalist models perform when given the same structural advantages as KBQA-R1.
> > >
> > > **Given the massive token consumption and prohibitive API costs associated with running multi-turn ReACT loops and complex schema navigation using proprietary frontier models, we evaluated these agents on a random subset of 500 queries (200 from GrailQA, 200 from WebQSP, and 100 from GraphQ).** The complete experimental results are presented in the table below:
> > >
> > > | Method | Model | WebQSP (F1) | GrailQA (F1) | GraphQ (F1) | Average (F1) | Avg Turns | AVG In/Out Tokens |
> > > | :--- | :--- | :--- | :--- | :--- | :--- | :--- | :--- |
> > > | **SPARQL ReACT** | GLM-5 | 0.486 | 0.638 | 0.379 | 0.525 | 7.10 | 10,792 / 1,952 |
> > > | *(Coding Agents)* | Kimi-K2.5 | 0.491 | 0.623 | 0.365 | 0.519 | 6.05 | 7,583 / 4,615 |
> > > | | gpt-5.3-codex | 0.545 | 0.656 | 0.427 | 0.566 | 5.20 | 4,994 / 454 |
> > > | | claude-sonnet-4-6 | 0.542 | 0.654 | 0.364 | 0.551 | 5.79 | 8,546 / 824 |
> > > | **KBQA-R1 Harness** | GLM-5 | 0.645 | 0.702 | 0.485 | 0.636 | 8.43 | 14,235 / 977 |
> > > | *(Policy Comparison)* | Kimi-K2.5 | 0.733 | 0.739 | 0.533 | 0.695 | 7.24 | 12,880 / 2,928 |
> > > | | gpt-5.3-codex | 0.711 | 0.763 | 0.515 | 0.693 | 3.43 | 4,537/ 374|
> > > | | claude-sonnet-4-6 | 0.767 | 0.763 | 0.518 | 0.716 | 5.45 | 15,325 / 2,959 |
> > > | **KBQA-R1 (Ours)** | Llama-3.1-8B | **0.842** | **0.856** | **0.554** | **0.790** | **2.91** | 3,994/ 328 |
> > >
> > > (*Note: The Kimi-K2.5 results reported here differ slightly from those in our previous rebuttal response, as the earlier figures were obtained on the full test set whereas all results in this table are evaluated on the consistent 500-query subset.*)
> > >
> > >
> > > These results reveal two key insights. First, our structured Harness consistently lifts all frontier models over their sparql-ReACT counterparts, confirming the value of the interaction framework itself. Second, and more importantly, even the best Harness-equipped frontier coding LLM(claude-sonnet-4-6, avg F1 0.716) falls short of KBQA-R1 (0.790), despite consuming far more tokens. This gap demonstrates that generalist models rely on costly test-time trial-and-error to discover valid schema paths, whereas KBQA-R1 internalizes the navigation policy through Referenced Rejection Sampling and RL training, achieving higher accuracy with fewer turns and lower token overhead.
> > >
> > > We commit to including these complete experimental results and the detailed discussion above in the camera-ready version of the paper to ensure a clear comparison between our specialized framework and generalist agentic systems. We promise that all supplementary experiments from the rebuttal period will be included in the camera-ready version if the paper is accepted.

---

### Decision · Program_Chairs · 2026-04-30

**Decision:**

Accept (regular)

**Comment:**

This paper presents KBQA-R1, a reinforcement learning framework that treats knowledge base question answering as a multi-turn decision process. The paper is technically sound and well-written. The rebuttal has addressed the concerns raised by reviewers. All reviewers are positive on the paper. I recommend this paper for Acceptance.